# Development and validation of a $NO_x^+$ ratio method for the quantitative separation of inorganic and organic nitrate aerosol using CV-UMR-ToF-ACSM

Farhan R. Nursanto[1], Douglas A. Day[2,3], Roy Meinen[4], Rupert Holzinger[4], Harald Saathoff[5], Jinglan Fu[5,6], Jan Mulder[6], Ulrike Dusek[6], and Juliane L. Fry[1]

[1]Meteorology and Air Quality, Environmental Sciences Group, Wageningen University and Research, 6708PB Wageningen, the Netherlands
[2]Cooperative Institute for Research in Environmental Sciences, University of Colorado Boulder, Boulder, CO, USA
[3]Department of Chemistry, University of Colorado Boulder, Boulder, CO, USA
[4]Department of Physics, Institute for Marine and Atmospheric Research Utrecht, Utrecht University, Princetonplein 5, 3584CC Utrecht, the Netherlands
[5]Institute of Meteorology and Climate Research, Karlsruhe Institute of Technology, Eggenstein-Leopoldshafen, Karlsruhe, Germany
[6]Centre for Isotope Research, Energy and Sustainability Research Institute Groningen, University of Groningen, 9747AG Groningen, the Netherlands

**Correspondence:** Farhan R. Nursanto (farhan.nursanto@wur.nl) and Juliane L. Fry (juliane.fry@wur.nl)

**Abstract.** Particulate nitrate is a major component of ambient aerosol around the world, present in inorganic form mainly as ammonium nitrate, and also as organic nitrate. It is of increasing importance to monitor ambient particulate nitrate, a reservoir of urban nitrogen oxides that can be transported downwind and harm ecosystems. The unit-mass-resolution time-of-flight aerosol chemical speciation monitor equipped with capture vaporizer (CV-UMR-ToF-ACSM) is designed to quantitatively

monitor ambient $PM_{2.5}$ composition. In this paper, we describe a method for separating the organic and ammonium nitrate components measured by CV-UMR-ToF-ACSM based on evaluating the $NO_2^+/NO^+$ ratio ($NO_x^+$ ratio). This method includes modifying the ACSM fragmentation table, time averaging, and data filtering. By using the measured $NO_x^+$ ratio of $NH_4NO_3$ and a plausible range of $NO_x^+$ ratio for organic nitrate aerosol, the measured particulate nitrate can be split into inorganic and organic fractions. Data pre-treatment filters concentrations of particulate nitrate below 0.6-2.0 µg m$^{-3}$, depending on the time

averaging. The method detection limit, when considering $\pm 10\%$ absolute uncertainty of organic nitrate fraction, is found to be 2 µg m$^{-3}$ (120 min averaging) to 10 µg m$^{-3}$ (10 min averaging) for total particulate nitrate concentration and 10% (120 min) to 20% (10 min) for organic nitrate fraction. We show that this method is able to distinguish periods with inorganic or organic nitrate as major components at a rural site in the Netherlands. A comparison to a high-resolution time-of-flight aerosol mass spectrometer equipped with a standard vaporizer (SV-HR-ToF-AMS) and positive matrix factorization (PMF) method shows

similar response of increasing particulate organic nitrate fraction with uncertainties mainly from sensitivity to fragmentation table correction when obtaining $NO_2^+$ signal. We propose that researchers use this $NO_x^+$ ratio method for CV-UMR-ToF-ACSM (adapting the appropriate fragmentation table and data pre-treatment for each specific application) to quantify the particulate organic nitrate fraction at existing monitoring sites in order to improve understanding of nitrate formation and speciation.

# 1 Introduction

In the current age of decreasing sulfur emissions, nitrate is becoming a principal aerosol component globally and regionally (Adams et al., 1999; Metzger, 2002; Liao et al., 2003; Rodriguez and Dabdub, 2004; Feng and Penner, 2007; Bauer et al., 2007; Paulot et al., 2016; Bian et al., 2017; Vasilakos et al., 2018; Drugé et al., 2019; Lu et al., 2021). In addition to an increasing aerosol fraction of ammonium nitrate ($NH_4NO_3$), ambient organic nitrates (ON) produced through the oxidation of volatile organic compounds (VOCs) in the presence of nitrogen oxides ($NO_x$) can condense into the particulate phase or grow new particles (Huang et al., 2019a, b; Song et al., 2024). The particulate ON (pON) contribution to total particulate nitrate mass ($pNO_3$) is substantial (Ng et al., 2017), with an average fraction of 17%-31% in China (Yu et al., 2024), 34%-44% in Europe (Kiendler-Scharr et al., 2016), and large differences between urban and rural areas (Fisher et al., 2016; Schlag et al., 2016; Romer Present et al., 2020; Yu et al., 2024). Improved understanding of pON fraction in different regions can provide insight into chemical mechanisms of secondary aerosol formation (Pye et al., 2015; Lee et al., 2016; Ng et al., 2017; Zare et al., 2018).

ON flux worldwide accounts ~25% of the total nitrogen deposition (Jickells et al., 2013). Zare et al. (2018) estimated, via WRF-Chem simulations in the southeast United States, that 60% of $NO_x$ loss is related to ON chemistry. Similar to inorganic nitrate, ON also can be regarded as a $NO_x$ reservoir, because thermal or photolysis processes can re-release $NO_x$. The partitioning between the gas-phase and particulate ON (Zare et al., 2018) can affect this reservoir lifetime, and thus the spatial scale of transport of urban nitrogen emissions from their source, determining how far downwind these emissions can harm natural habitats (Fields, 2004; Bobbink and Hicks, 2014; Erisman et al., 2015; Melillo, 2021).

The $NO_x^+$ ratio method, first described by Farmer et al. (2010), is a robust method to separate the total $pNO_3$ signal measured by high resolution-aerosol mass spectrometers (AMS) into particulate ammonium nitrate (pAmN) and particulate organic nitrate (pON) using the variation of $NO_2^+/NO^+$ ion ratios (subsequently referred to as $NO_x^+$ ratios) in the mass spectra observed. This method has been successfully used to analyze pON composition in several studies (Fry et al., 2013; Pye et al., 2015; Kiendler-Scharr et al., 2016; Ng et al., 2017; Fry et al., 2018; Huang et al., 2019a, b; Brownwood et al., 2021; Day et al., 2022a).

The basis of the $NO_x^+$ ratio method comes from the different fragmentation patterns of chemical species due to the interaction of the mass spectrometer's vaporizer and ionizer with the analytes. The empirical observation shows that nitrates attached to an organic moiety have different fragmentation patterns compared to nitrate in the form of $NH_4NO_3$, and also other less volatile inorganic nitrate. Thus, each nitrate will have different $NO_x^+$ ratios, $R_\nu$, as shown below in Eq. 1 (Day et al., 2002; Francisco and Krylowski, 2005; Farmer et al., 2010; Drewnick et al., 2015; Hu et al., 2016b; Day et al., 2022a).

$$R_\nu = \frac{(C_{NO_2^+})_\nu}{(C_{NO^+})_\nu} \tag{1}$$

$\nu$: nitrate compound or mixture measured

$C_{NO_2^+}$: signal intensity of $NO_2^+$

$C_{NO^+}$: signal intensity of $NO^+$

The $NO_x^+$ ratio of the observed air ($R_{obs}$) falls between the $NO_x^+$ ratios of pure pAmN ($R_{pAmN}$) and pure pON ($R_{pON}$). The time-varying mass fraction of particulate organic nitrate ($f_{pON}$, referring to pNO$_3$ existing as pON), and particulate ammonium nitrate ($f_{pAmN}$, referring to pNO$_3$ existing as pAmN) can be extracted from this time-varying $R_{obs}$ using Eqs. 2 and 3 (Farmer et al., 2010).

$$f_{pON} = \frac{(R_{obs} - R_{pAmN})(1 + R_{pON})}{(R_{pON} - R_{pAmN})(1 + R_{obs})} \tag{2}$$

$$f_{pAmN} = 1 - f_{pON} \tag{3}$$

The aerosol chemical speciation monitor (ACSM; Aerodyne Inc.) is a unit-mass resolution (UMR) mass spectrometry instrument intended for continuous ambient aerosol monitoring (Ng et al., 2011; Fröhlich et al., 2013), unlike its predecessor, the aerosol mass spectrometer (AMS; Aerodyne Inc.) which is designed primarily for research (Drewnick et al., 2005). In this work, we explored whether the ACSM can be used to determine pAmN and pON in the same way as has been successfully demonstrated for the AMS. ACSMs are used extensively in monitoring networks, such as the sites in the Aerosol, Clouds and Trace Gases Research Infrastructure (ACTRIS) network in Europe (https://www.psi.ch/en/acsm-stations/overview-full-period, last access: 6 November 2024) and the Atmospheric Science and Chemistry mEasurement NeTwork (ASCENT) network in the USA (https://ascent.research.gatech.edu, last access: 6 November 2024).

For monitoring purposes, a capture vaporizer (CV) and an intermediate pressure lens (IPL) are recommended by Aerodyne for improved quantification of the PM$_{2.5}$ fraction, relative to a standard vaporizer (SV) and standard lens (Zheng et al., 2020). Almost half of the ACSMs in the ACTRIS network in Europe use CV. The CV is designed is to increase particle collision events with the vaporizer surface by having a narrow entrance, resulting in a particle collection efficiency (CE) of 1 and better mass closure of PM$_{2.5}$ monitoring (Jayne and Worsnop, 2016; Hu et al., 2017; Xu et al., 2017; Liu et al., 2024). The enhanced thermal decomposition, however, shifts the fragmentation pattern toward smaller ion fragments (Hu et al., 2017, 2018a; Xu et al., 2017; Zheng et al., 2020). Therefore, the $NO_x^+$ ratio is substantially lower with CV compared to SV due to favored $NO^+$ formation. In consequence, the $NO_x^+$ ratio method's applicability in CV-based measurements is limited by the $NO_2^+$ detection limit (<0.1 µg m$^{-3}$; Hu et al. (2017)).

While high resolution mass spectrometers can separate non-$NO_x^+$ peaks which are detected at the same nominal $m/z$ (mass-to-charge ratio) as $NO_x^+$ peaks (30 for $NO^+$ and 46 for $NO_2^+$), UMR analysis requires estimations based on related ions at other $m/z$. These estimations are incorporated into data workup by the implementation of a fragmentation table, which subtracts an estimated amount of organic at $m/z$ 30 and $m/z$ 46, based on the signal at another related organic-only $m/z$. The default fragmentation table typically applied for the analysis of UMR spectra is based on generalized fragment mass composition of ambient aerosol composition measured using SV-based instruments (Allan et al., 2004; Ulbrich et al., 2009), and thus not suitable for CV-UMR-ToF spectra that have different fragmentation patterns. Using a CV-HR-ToF-AMS, Hu et al. (2017) determined the organic fragment interference to $NO^+$ in $m/z$ 30 and to $NO_2^+$ in $m/z$ 46 for CV-UMR measurements in a biogenically-dominated dataset, but no study has yet shown this calculation adapted to general ambient aerosol composition.

This work aims to adapt the $NO_x^+$ method to separate pAmN and pON signals to CV-UMR-ToF-ACSM measurements. We first provide a revised fragmentation table for $m/z$ 30 and $m/z$ 46 compatible with CV-UMR-ToF-ACSM measurements with

varying composition to better calculate $NO^+$ and $NO_2^+$ signal contributions. Second, we show the variation of experimental $NO_x^+$ ratio for pAmN in CV-UMR-ToF-ACSM instruments and determine the $NO_x^+$ ratio for pON. Third, we demonstrate the capability of data pre-treatments (filtering and time averaging) to overcome the low and noisy ratio signals produced by CV in ambient measurements. Fourth, the proposed $NO_x^+$ ratio method is applied to an extended ambient dataset (at the Cabauw site of the Ruisdael Observatory Network) to test its robustness for changing ambient aerosol mixtures. Lastly, the formation of pAmN and pON in a chamber experiment measured using CV-UMR-ToF-ACSM is used for method validation by comparing results with SV-HR-ToF-AMS.

## 2 Instrumentation

### 2.1 Description of ToF-ACSM

A ToF-ACSM (Aerodyne Inc.) is the main instrument used in this study, allowing the chemical analysis of non-refractory organics (Org), ammonium ($NH_4$), nitrate ($NO_3$), sulfate ($SO_4$), and chloride (Chl) in the aerosol phase (Ng et al., 2011; Fröhlich et al., 2013). In comparison to compact time-of-flight (cToF)-AMS (Drewnick et al., 2005) and HR-ToF-AMS (DeCarlo et al., 2006), ToF-ACSM is more compact in size, lower in price and operational cost, simpler in analysis, and requires less user intervention, which makes this instrument practical for long-term monitoring but still comparable to the AMS (Fröhlich et al., 2013). ToF-ACSM uses a three-way valve system that allows automatic switching between the sample and filter mode, unlike ToF-AMS, which has a mechanical chopper that physically blocks the particle beam. The lack of a chopper (but with the use of particle time-of-flight chamber) in the ToF-ACSM, however, removes the particle sizing feature, which makes it similar to the quadrupole-ACSM (Q-ACSM) but with better mass resolution and detection limits.

To conduct the various analyses in this paper, we primarily use data from two ToF-ACSM instruments with identical setup. The instruments are managed by Utrecht University (UU) and University of Groningen (RUG), part of a larger monitoring network of Ruisdael Observatory in the Netherlands (https://ruisdael-observatory.nl, last access: 6 November 2024). We label the instruments as ACSM-UU and ACSM-RUG. The instrument setup for ambient measurements uses a combination of a $PM_{2.5}$ size-cut cyclone, an intermediate-pressure lens ($PM_{2.5}$ aerodynamic lens), and a capture vaporizer (CV, temperature $\sim$525 °C, Jayne and Worsnop (2016)) that has been aligned with the particle beam. Together, they configure the ToF-ACSM as a $PM_{2.5}$ monitor (Xu et al., 2017) with unit mass resolution. The instrument provides UMR mass spectra with default 10 $min$ time resolution, analyzed using Tofware v3.3 in Igor Pro 8. The fractions of the UMR signal are assigned to different aerosol species using the fragmentation table.

### 2.2 Ambient measurements with ACSM

We use an ambient dataset measured using ACSM-UU deployed in Cabauw, the Netherlands, for method development and case studies. The ambient data were measured between 18 April 2023 to 15 April 2024 with some gaps (net 205 days of data) as part of the continuous monitoring of the Ruisdael Observatory network. Aerosol measurements were carried out

with an inlet height of 4.5 m above the ground at the Cabauw tower (51.97 °N, 4.93 °E), an infrastructure of the Royal Netherlands Meteorological Institute (KNMI, the Netherlands, https://www.knmi.nl/home, last access: 9 November 2024). The site is surrounded by agricultural lands in the province of Utrecht, the Netherlands, a relatively nitrogen-polluted rural site.

Ambient air is sampled through a stainless-steel inlet system with a $PM_{2.5}$ size-cut cyclone (URG-2000-30ED) and a Nafion dryer with a sampling flow rate of $\sim$2 L min$^{-1}$, of which in average 1.23 cm$^3$ s$^{-1}$ (0.07 L min$^{-1}$) is sampled by the ACSM. The calibrations of ionization efficiency (IE) and relative IE (RIE) were performed using 300 nm particles from ammonium nitrate ($NH_4NO_3$) and ammonium sulfate (($NH_4)_2SO_4$)) solutions (size-selected with a differential mobility analyzer, model TSI 3081 and co-sampled with a condensation particle counter, model TSI 3750). The average IE value for the instrument is 169 ions pg$^{-1}$ for $NO_3$, and RIE values are 1.40, 1.58, 1.30, and 3.37 for Org, $SO_4$, Chl, and $NH_4$, respectively. RIE's used for Org and Chl were not measured, and instead applied as default values, as is common practice. The detection limits at 10 min time resolution are 0.38 µg m$^{-3}$ for Org, 0.12 µg m$^{-3}$ for $NH_4$, 0.07 µg m$^{-3}$ for $NO_3$, 0.07 µg m$^{-3}$ for $NO^+$ ($m/z$ 30), 0.04 µg m$^{-3}$ for $NO_2^+$ ($m/z$ 46), 0.11 µg m$^{-3}$ for $SO_4$, and 0.09 µg m$^{-3}$ for Chl.

## 2.3 Chamber measurements with ACSM and AMS

ACSM-RUG was deployed to measure aerosol in chamber experiments conducted in Aerosol Interaction and Dynamics in the Atmosphere (AIDA) chamber, a facility maintained by Karlsruhe Institute of Technology (KIT), Germany. Chamber experiments were conducted in 2023 and 2024 as part of the Cloud-Aerosol Interactions in a Nitrogen-dominated Atmosphere (CAINA) project (https://sites.google.com/view/cainaproject/, last access: 6 November 2024).

Chamber air is sampled using stainless steel tubing equipped with a Nafion dryer and a sampling flow of $\sim$2 L min$^{-1}$ of which in average 1.44 cm$^3$ s$^{-1}$ (0.09 L min$^{-1}$) is sampled by the ACSM. The average IE value for the instrument is 152 ions pg$^{-1}$ for $NO_3$. The ACSM instrument is run with 2 min time resolution unlike the default setting to capture more variation in the aerosol composition. The detection limits at 2 min time resolution for the ACSM-RUG instrument are 0.20 µg m$^{-3}$ for Org, 0.19 µg m$^{-3}$ for $NH_4$, 0.17 µg m$^{-3}$ for $NO_3$, 0.17 µg m$^{-3}$ for $NO^+$ ($m/z$ 30), 0.03 µg m$^{-3}$ for $NO_2^+$ ($m/z$ 46), 0.02 µg m$^{-3}$ for $SO_4$, and 0.05 µg m$^{-3}$ for Chl.

In addition, a high-resolution time-of-flight aerosol mass spectrometer (HR-AMS, Aerodyne Research Inc.) from KIT is connected to the chamber via a 6 mm (4 mm internal diameter) stainless steel tube. The instrument is equipped with a $PM_{2.5}$ aerodynamic lens to measure the non-refractory $PM_{2.5}$ components, at a time resolution of 1 min (DeCarlo et al., 2006; Canagaratna et al., 2007; Williams et al., 2013), averaged to 2 min for this comparison. The operation of the AMS is explained in previous publications (Huang et al., 2019a; Song et al., 2022). Briefly, chamber air is sampled with a flow of 1.08 L min$^{-1}$, of which in average 84 cm$^3$ min$^{-1}$ is sampled by the AMS (Gao et al., 2022). The aerosol particles are then focused into a narrow beam by a $PM_{2.5}$ aerodynamic lens with an effective complete transmission for particle sizes ranging from 70 to 2500 nm (vacuum aerodynamic diameter; $D_{va}$) and heated by a standard vaporizer at 600 °C. The resulting vapors are ionized by electron impact (70 eV) and characterized by a time-of-flight mass spectrometer. The AMS ionization efficiency is calibrated using 300 nm dried $NH_4NO_3$ aerosol particles to give an average IE $NO_3$ of 185.0 ions pg$^{-1}$. The AMS data are analysed using the software packages Squirrel 1.66E and PIKA 1.26E in Igor Pro 8. To account for the effect of particle bouncing loss,

chemical-composition-based collection efficiency (0.5) are applied to calculate the particle mass concentration (Middlebrook et al., 2012).

# 3   Development of a fragmentation table for CV-UMR-ToF-ACSM from AMS spectral database and ACSM chamber experiment spectra

## 3.1   General fragmentation table for typical ambient dataset

In the UMR-ACSM instruments, ions detected at $m/z$ 30 and $m/z$ 46 can originate both from nitrate ($NO^+$ and $NO_2^+$) and organic fragments. However, it is known that some fragments produced by processes in the vaporizer and ionizer can be related to one another (Allan et al., 2004). For instance, UMR peaks at $m/z$ 29, $m/z$ 42, $m/z$ 43, and $m/z$ 45 are mainly the product of further fragmentation of fragments at $m/z$ 30 and $m/z$ 46 and assumed to be exclusively of organic origin. The aim of the fragmentation table, with respect to the $NO_x^+$ species, is to predict the signal contribution of organic fragments at $m/z$ 30 and
$m/z$ 46 based on the masses measured at $m/z$ 29, $m/z$ 42, $m/z$ 43, and $m/z$ 45, and subsequently to extract the signal that can be attributed to $NO^+$ and $NO_2^+$.

Our starting point is the default fragmentation table from Allan et al. (2004) (see Table 1). A fragmentation table consists of columns dividing the raw mass spectra into chemical species, with rows denoting entries for different UMR nominal masses. Thus, each entry consists of components which are added up to obtain the species concentration at a specific nominal mass.
These components can be the whole peak of an $m/z$ $x$, referred to as an integer number (with square brackets in this paper, "[$x$]"), or the contribution of a certain species to $m/z$ $x$ in the fragmentation table, denoted as "frag_species[$x$]." A multiplier $a$ (positive or negative) is included if the addition or subtraction of the component is fractional. The fragmentation table for ToF-AMS and ToF-ACSM are identical, except that gas-phase species contribution must also be removed in ToF-AMS. For the ACSM, due to the automatic filter sampling cycle and subtraction, gas-phase species are already removed. The fragmentation
table developed in this paper, therefore, is applicable to a CV-UMR aerosol mass spectrometer. Because the training data set incorporated multiple chamber and ambient measurements with different instruments, it should be applicable for a range of typical measurement configurations, but users should be aware of the potential effects of the instrument condition (e.g., vaporizer temperature, particle beam alignment, measurement history).

In the default fragmentation table (which was developed using an SV-based instrument), the signal at $m/z$ 46 is assigned
exclusively to $NO_2^+$, and the relationship of organic signal at $m/z$ 30 is found to be only 0.022 times the magnitude of organic signal at $m/z$ 29. Switching from SV to CV modifies the signal ratio between organic and inorganic fragments at $m/z$ 30 and $m/z$ 46, because of greater organic fragmentation in CV (Hu et al., 2018a). It also leads to greater nitrate fragmentation and consequently smaller $NO_2^+$ signal, which makes organic contribution at $m/z$ 46 more important. For instance, Fry et al. (2018) found larger contributions of organic fragments at $m/z$ 30 and $m/z$ 46 than the default fragmentation table in a semi-polluted
biogenically-influenced air analyzed with an SV-HR-ToF-AMS. Therefore, modifications to frag_NO3[46] and frag_NO3[30] entries (later referred to as $C_{NO_2^+}$ and $C_{NO^+}$ to calculate $NO_x^+$ ratio outside fragmentation table context) must be established for CV-based instruments.

**Table 1.** Excerpt of fragmentation table for Org and $NO_3$ species in $m/z$ 30 and $m/z$ 46. Second and third column shows entries originated from the default fragmentation table of Allan et al. (2004) (used in Tofware v3.3). Fourth and fifth column shows entries proposed to develop revised CV-UMR-ToF-ACSM fragmentation table in this study.

| $m/z$ | Allan et al. (2004), default fragmentation table | | Proposed for general CV-ToF-ACSM | |
| --- | --- | --- | --- | --- |
| | Org | $NO_3$ | Org | $NO_3$ |
| 30 | $0.022 \cdot$ frag_Org[29] | [30], -frag_Org[30] | $a_{\text{Org}[30],[i]} \cdot$ frag_Org[$i$] [(a)] | [30], -frag_Org[30] |
| 46 | - | [46] | $a_{\text{Org}[46],[i]} \cdot$ frag_Org[$i$] [(b)] | [46], -frag_Org[46] |

$i$ represents UMR masses tested against $m/z$ 30 and $m/z$ 46 in this study, which includes frag_Org[29], frag_Org[42], frag_Org[43], and frag_Org[45]. See the list in the footnote of Table S3 of SI.

(a) $a_{\text{Org}[30],[i]}$ is the multiplier for frag_Org[30] component, obtained from the slope of ODR fit between frag_Org[30] and frag_Org[$i$].

(b) $a_{\text{Org}[46],[i]}$ is the multiplier for frag_Org[46] component, obtained from the slope of ODR fit between frag_Org[46] and frag_Org[$i$].

To make a revised fragmentation table applicable for general ambient organic aerosol (OA) mixtures, a variety of organic aerosol profiles is necessary. We use 25 CV-HR-ToF-AMS spectra (including both nitrates and non-nitrate organics) from the AMS spectral database (http://cires1.colorado.edu/ jimenez-group/AMSsd_CV, last access: 6 November 2024) and 6 CV-UMR-ToF-ACSM spectra from chamber experiments. The CV-HR-ToF-AMS database mass spectra include 3 chamber experiments, 7 factors of positive matrix factorization (PMF) analysis from ambient measurements, and 15 laboratory standards measurements (Hu et al., 2017; Carlton et al., 2018; Hu et al., 2018a, b), summarized in Table S1 in the Supplementary Information (SI). The CV-UMR-ToF-ACSM mass spectra were measured from experiments conducted in the AIDA chamber (see Section 2.3). These spectra were obtained using vaporizer temperature ranging from 525 to 600 $^{\circ}$C (see Table S1 and S2). Therefore, the revised fragmentation table should be valid for CV-based instruments run in this temperature range.

Using these data, we determine the multipliers $a$ used in a revised calculation of frag_Org[30] and frag_Org[46] (see Table 1, fourth and fifth column and Table S1, third and fourth column). The new multipliers are determined by performing orthogonal distance regression (ODR) constrained to a zero intercept of mass spectra in UMR. For HR-AMS spectra, the dataset is "degraded" from HR into UMR spectra by summing HR Org fragments to their respective nominal mass in each AMS spectrum. Note that for UMR-ACSM spectra, because we cannot separate species at the same nominal $m/z$, we only use chamber experiments that are assumed nitrate-free or to contain negligible nitrate (no seed or precursor for inorganic and organic nitrate). Therefore, all signals at $m/z$ 30 and $m/z$ 46 are exclusively organic fragments. We perform ODR fits of frag_Org[30] and frag_Org[46] against a list of chemically related masses (frag_Org[i]). The slope of the ODR fits to determine the multiplier $a_{\text{Org}[30],[i]}$ and $a_{\text{Org}[46],[i]}$ are summarized in Table S3, alongside the list of fragments of $i$ that contribute to each nominal mass, in the footnote. It is found that frag_Org[30] is best correlated with frag_Org[29] (see Table S3), where $a_{\text{Org}[30],[29]} = 0.311 \pm 0.016$ (mean $\pm$ uncertainty, $r^2 = 0.88$, see Fig. 1a). On the other hand, frag_Org[46] has the best correlation with frag_Org[45] (see Table S3), where $a_{\text{Org}[46],[45]} = 0.305 \pm 0.037$ (mean $\pm$ uncertainty, $r^2 = 0.43$, see Fig. 1b). The final revised fragmentation table in CV-UMR-ToF-ACSM is summarized in the conclusions (see Table 4).

We apply these new multiplier values to the full dataset and compare the results with those from multipliers described in Allan et al. (2004), the SI of Fry et al. (2018), and the SI of Hu et al. (2017) (see in Table S4). The result suggests that the

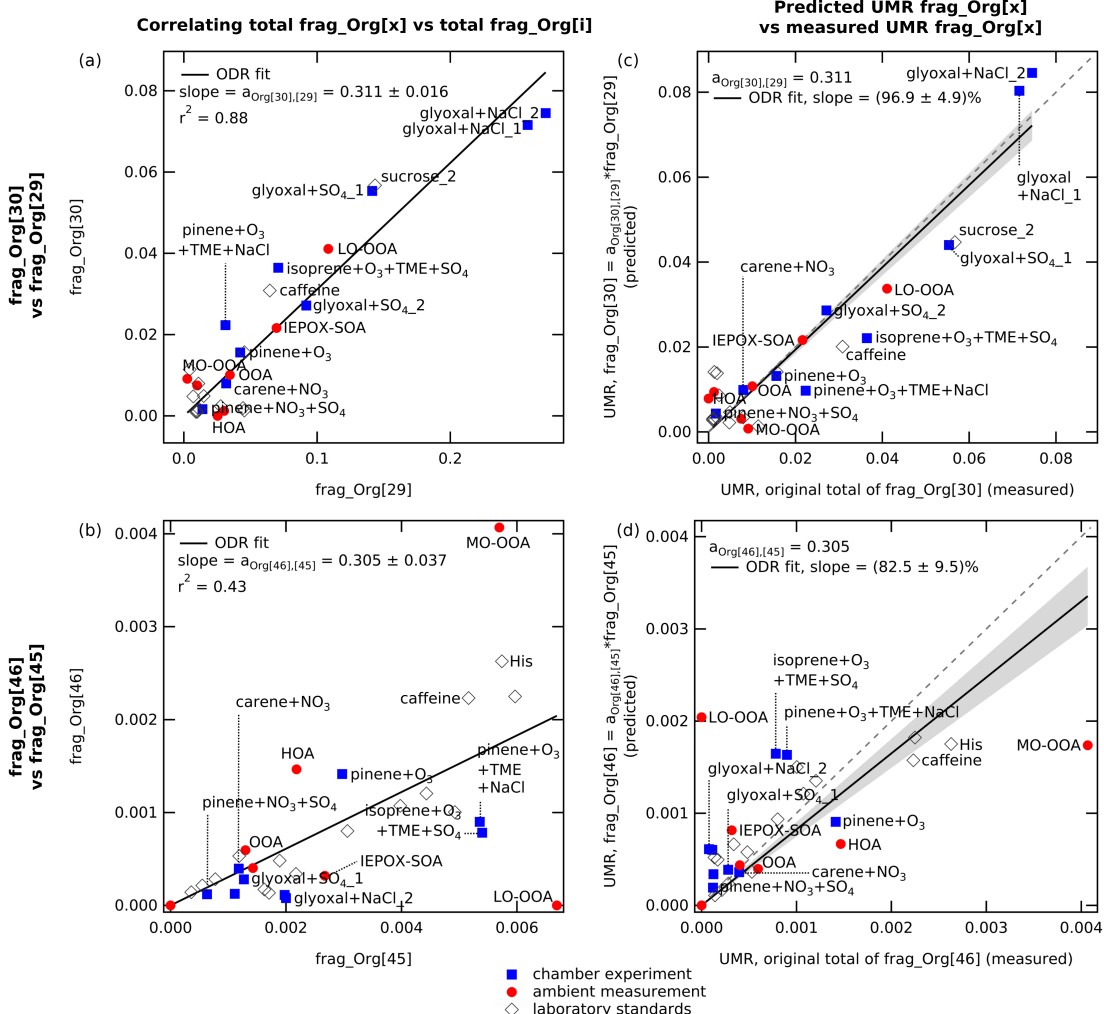

**Figure 1.** The left-hand panels show the best ODR fits (set to zero intercept) which are found in the relationship between the signal contributions of (a) frag_Org[30] vs frag_Org[29], and (b) frag_Org[46] vs frag_Org[45]. The correlations of all mass pairs are summarized in Table S3. The right-hand panels show the predicted organic contributions (based on the new multipliers) at each $m/z$ versus the measured amount. Plot (c) shows the predicted UMR frag_Org[30] against the measured total Org fragments in $m/z$ 30, and plot (d) shows the predicted UMR frag_Org[46] against the measured total Org fragments in $m/z$ 46. The figure demonstrates that the predicted frag_Org[$x$] slightly underestimates (slope = 0.83) but approached the measured frag_Org[$x$].

multiplier $a_{\text{Org}[46],[45]}$ determined here gives the best predicted frag_Org[46] over multipliers from other studies (see Table S4 third and seventh columns in the SI). Meanwhile, the multiplier $a_{\text{Org}[30],[29]}$ determined here performs similarly with the multipliers obtained from Hu et al. (2017) (SV and CV) for a dataset dominated by biogenic secondary organic aerosol (SOA). The plot of predicted UMR frag_Org[30] against the measured total Org fragments in $m/z$ 30 (see Fig. 1c) shows that the

multiplier determined here is able to estimate on average $96.9 \pm 4.9\%$ of the measured frag_Org[30]. Meanwhile, the similar plot for frag_Org[46] against the measured total Org fragments in $m/z$ 46 (see Fig. 1d) shows that the multiplier determined here estimates on average $82.5 \pm 9.5\%$ of the measured frag_Org[46]. The low signal intensity of both $m/z$ 46 and $m/z$ 45 may cause this underestimation and suggests that frag_Org[46] and frag_Org[45] may have a more complicated relationship; their correlation may vary substantially depending on the aerosol mixture. Therefore, it may be appropriate to modify the entry for frag_NO$_3$[46] according to the type of aerosol mixture analyzed. On the other hand, for complex ambient mixtures, the ensemble composition may produce spectra that are more similar to the average determined here. Analysis of CV-HR spectra from a variety of ambient samples would be required to determine the actual variation of organic contributions at $m/z$ 46.

### 3.2 Composition-specific fragmentation table

The multipliers determined in Section 3.1 are designated for typical ambient aerosol composition. In some cases, a composition-specific fragmentation table may be more appropriate to use. For instance, from a field study with biogenically-dominated composition, Hu et al. (2017) reported $a_{\text{Org}[30],[29]} = 0.32$ and $a_{\text{Org}[46],[45]} = 0.68$. By using selected chamber experiments in the same dataset, we can explore different multipliers that are compatible for different composition profiles. For chamber experiments that involves glyoxal (and its oligomers), we obtain $a_{\text{Org}[30],[29]} = 0.291 \pm 0.022$ (mean $\pm$ uncertainty, $n = 4$, $r^2 = 0.90$) and $a_{\text{Org}[46],[45]} = 0.082 \pm 0.036$ (mean $\pm$ standard deviation, $n = 4$, $r^2 = 0.35$). For chamber experiments that involve only terpenes (e.g., isoprene, $\alpha$-pinene), we obtain $a_{\text{Org}[30],[29]} = 0.476 \pm 0.067$ (mean $\pm$ uncertainty, $n = 5$, $r^2 = 0.82$) and $a_{\text{Org}[46],[45]} = 0.204 \pm 0.055$ (mean $\pm$ uncertainty, $n = 5$, $r^2 = 0.33$) which can be applied for chamber experiments with terpene as precursor. The ODR fit plots are given in Fig. S1. The multipliers to revise the fragmentation table in CV-UMR-ToF-ACSM for specific aerosol composition subsets is summarized in the conclusions (see Table 4).

## 4 Determination of NO$_x^+$ ratios for the CV-UMR-ToF-ACSM

### 4.1 NO$_x^+$ ratio of pure pAmN

To quantify the inorganic NO$_x^+$ ratio typical value and variability produced by the CV, we use repeated measurements from regular pAmN calibration from the two ACSM instruments described in Section 2. The $R_{\text{pAmN}}$ of ACSM-UU is found to be $0.0237 \pm 0.0009$ (mean $\pm$ uncertainty, $n = 5$), and $R_{\text{pAmN}}$ of ACSM-RUG is $0.0115 \pm 0.0002$ (mean $\pm$ uncertainty, $n = 3$). The values are similar but lower than other studies where $R_{\text{pAmN}}$ with CV was found to be 0.04-0.07 (Hu et al., 2016a), and all are $\sim$10 times lower than the typical $R_{\text{pAmN}}$ measured with the SV, 0.3-0.7 (Day et al., 2022b). We found that $R_{\text{pAmN}}$ values are very consistent for each instrument over time. A summary of $R_{\text{pAmN}}$ values and regression fit parameters from each measurement can be found in Table S5 in the SI.

Hu et al. (2017) found that the NO$_x^+$ ratio is affected by the aerodynamic lens alignment. The influence is greater in the CV since the vaporizer opening diameter is $\sim$2.5 mm (SI of Hu et al. (2018b)), smaller than that of SV, which is $\sim$3.8 mm (Drewnick et al., 2005). For optimum particle detection, the aerodynamic lens must point the particle beam at the center of

the vaporizer. Directed into the center, the particles enter the CV cavity and experience augmented thermal decomposition, at which the $NO_2^+$ signal intensity is at its minimum, while the $NO^+$ signal intensity is highest. The $NO_2^+$ signal intensity increases as the particle beam moves closer to the edge of the vaporizer, where the thermal decomposition is not as extensive as in the center, resembling how the SV works. Thus, it is important to consider performing lens alignment to obtain the correct result for the $NO_x^+$ ratio method. Monitoring the behavior of $m/z$ 30 and $m/z$ 46 in during pAmN calibration is a good way to determine whether the aerodynamic lens is well aligned. Combining the result of this study and the $R_{pAmN}$ range in Hu et al. (2017), value of $R_{pAmN}$ in the range of 0.01-0.07 can be used as reference to indicate properly aligned lens. Plotting the profile of the $NO_x^+$ ratio with movement in both the vertical and horizontal directions obtained during the alignment is the best diagnostic.

## 4.2 $NO_x^+$ ratio of pure pON

To obtain a $NO_x^+$ ratio for pON, one would ideally measure pure atmospherically-relevant ON, prepared through synthesis or in a chamber experiment. Often, such standards are not available and therefore experimental $R_{pON}$ cannot be easily determined for each instrument. To overcome this challenge, Day et al. (2022a) used the strategy of a "ratio-of-ratios" ($RoR = R_{pAmN}/R_{pON}$) which can be used to calculate $R_{pON}$ for any arbitrary instrument from its routinely-measured $R_{pAmN}$, and an average value for the RoR measured across many instruments under varying conditions. Day et al. (2022a) found that based on the relationship between $R_{pAmN}$ and $R_{pON}$ over a large range of measurement conditions, SV produces RoR of $2.75 \pm 0.70$ (mean $\pm$ 25% uncertainty).

There have not been enough studies yet that have determined $R_{pON}$ values in CV-based instruments in order to determine a robust $RoR$ estimate for CV instruments. In the work of Hu et al. (2017), a chamber experiment producing pure pON yielded $R_{pON} = 0.0045$, and with comparison to $R_{pAmN} = 0.06$ of their instrument, the value $RoR = 13.3$ is obtained. With $R_{pAmN}$ being 0.01 in CV as found in this study, using this $RoR$ of 13.3, $R_{pON}$ would be 0.0008 (approaching zero).

Similar to the approach of Kiendler-Scharr et al. (2016) that used the minimum measured value of $NO_x^+$ ratio to set a fixed $R_{pON}$ value, to estimate $R_{pON}$, we use the lowest measured $NO_x^+$ ratio from a chamber experiment in which we expected to produce pON, with no inorganic nitrate present. The selected experiment used glyoxal as SOA precursor, $NO_2$ and $O_3$ to produce $NO_3$ radical as the major oxidant, and sodium chloride (NaCl) seed to form SOA containing organic nitrates at 90% relative humidity. This experiment was conducted in the AIDA chamber at IMK KIT, Germany as part of CAINA project. For the spectra analysis, we use the fragmentation table specific for glyoxal-related chamber experiment, as described in Section 3.2.

While this experiment should produce pure organic nitrate aerosol, during pON formation, we observe an increase in $NH_4$ which could happen for two reasons. First, ammonium nitrate impurities can be formed from reactions or chamber wall repartitioning. Second, particulate water can be incorrectly assigned as $NH_4$ through fragmentation table correction at the $m/z$ 16 and $m/z$ 17 (see complete fragmentation table in Allan et al. (2004)). Both can result in a higher $NO_x^+$ ratio than as we would have if the total nitrate were purely pON. Therefore, we derive two bounding $R_{pON}$ values from the glyoxal chamber experiment. First, we determine the $R_{pON}$ from the experiment by assuming pure pON formation to obtain an upper limit. Second, to

obtain the lower limit of $R_{pON}$, we assume all observed $NH_4$ increase is $NH_4NO_3$ aerosol and subtract this equivalent amount of inorganic nitrate, with the inorganic $NO_x^+$ ratio, from the total $NO_3$, in order to obtain a lower limit of pON time series and calculate the $R_{pON}$. This strategy has been described as the "excess $NH_4$" method in Takeuchi and Ng (2019). By rearranging the Eq. 2 to Eq. S2, we can obtain $R_{pON}$ using $f_{pON}$, $R_{pAmN}$, and $R_{obs}$ for the lower limit experiment (see details in Section S3 of SI).

The upper limit experiment gives $R_{pON} = 0.0035$ (see Fig. 2a). If we compare to $R_{pAmN} = 0.0115 \pm 0.0002$ (measured separately with pure AmN), $R_{pON} = 0.0035$ obtained from the upper limit experiment gives $RoR = 3.29$ (see Fig. 2a). This value is higher than $RoR$ for SV-AMS but lower than $RoR$ obtained from Hu et al. (2017).

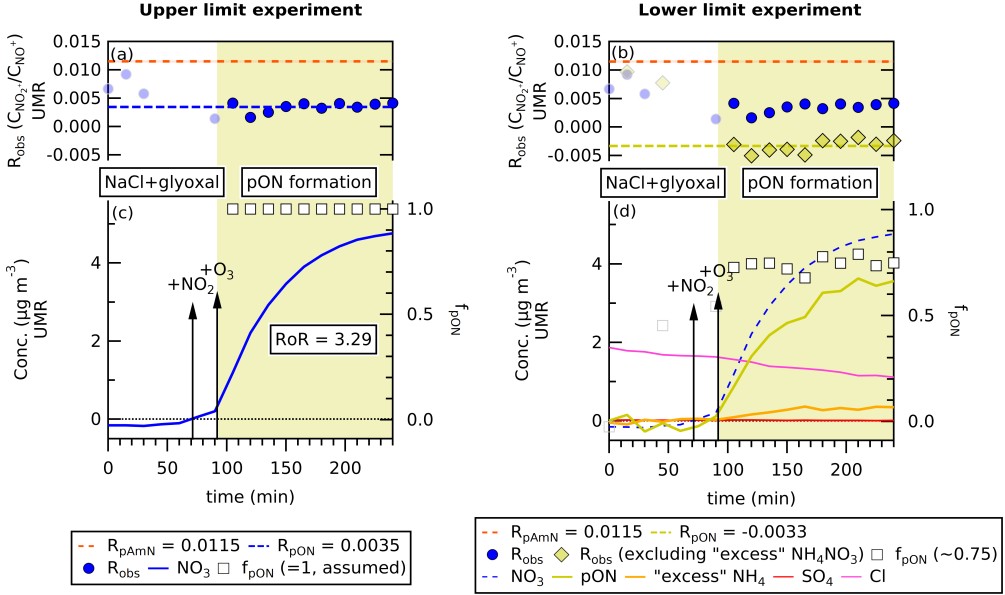

**Figure 2.** The time series of (a and b) $R_{obs}$, (c and d) ACSM species concentration (in µg m$^{-3}$, left bottom axis), and $f_{pON}$ (right bottom axis) of glyoxal+$NO_3$ chamber experiment at 15 min time averaging. The UMR fragmentation table specific for glyoxal is used to obtain $C_{NO^+}$ and $C_{NO_2^+}$. Panels (c) and (d) shows the progression of $NO_3$ concentration, compared to panels (a) and (b) for the $NO_x^+$ ratio during the formation of pON. By averaging $R_{obs}$ after pON formation started, the left-hand panel shows that $R_{pON} = 0.0035$ is obtained when assuming $f_{pON} = 1$. When possible inorganic impurity is removed by assuming "excess" $NH_4$ are $NH_4NO_3$, the average $f_{pON}$ is found to be 0.75. The excess $NH_4$ concentration is obtained by subtracting total $NH_4$ concentration by the average $NH_4$ concentration before the addition of $NO_2$ and $O_3$ (to exclude any possible $NH_4Cl$ contribution). By calculating $R_{pON}$ using the obtained $f_{pON}$ from "excess $NH_4$" method, $R_{pAmN}$ and $R_{obs}$, the value $R_{pON} = -0.0033$ is obtained, suggesting an overcorrection (see text). The value $R_{pON} = 0.0035$ from the lower limit experiment and $R_{pAmN} = 0.0115 \pm 0.0002$ (mean $\pm$ uncertainty) give a ratio-of-ratios ($RoR = R_{pAmN}/R_{pON}$) of 3.29.

On the other hand, the lower limit experiment gives $R_{pON} = -0.0033$ (see Fig. 2b). A negative (or below zero) $R_{pON}$ value is not chemically possible for the ratio. This value indicates an overcorrection, or that $R_{pON}$ is varying around the zero value

(small positive and negative) when the air mixture is strictly inorganic nitrate free. Thus, for calculation purposes, we use $R_{pON}$ = 0.0001 to represent the smallest possible $NO_x^+$ ratio for CV-ACSM measurements.

As the $RoR$ from the experiments (including Hu et al. (2017), $RoR$ = 13.3) are very different, we set the upper limit of $R_{pON}$ to be $R_{pON}$ obtained using $RoR$ = 3.29, and the lower limit to be $R_{pON}$ = 0.0001 in CV. The calculated $R_{pON}$ for two CV-ToF-ACSM deployed in this study are summarized in Table 2. These limits are used to determine the uncertainty of $f_{pON}$ calculation. Since we see the tendency of $m/z$ 46 signal intensity (and thus $NO_2^+$) to be produced in the vaporizer in relatively small quantities compared to $NO^+$, the tendency of $R_{pON}$ therefore is also to approach zero (non-normal distribution). With only limited information about $R_{pON}$ in CV unlike SV, we use the geometric mean instead of arithmetic mean to establish the expected central value of $R_{pON}$. We note that it is common to use geometric means to eastimate averages of ratios.

**Table 2.** Summary of measured $R_{pAmN}$ (including uncertainties) and calculated $R_{pON}$ (upper and lower limits as determined from pON in glyoxal+$NO_3$ chamber experiment). The geometric mean is considered as central value since $R_{pON}$ is likely approaching zero. Values are for ACSM-UU (employed for ambient measurements in Cabauw) and the ACSM-RUG (employed for AIDA chamber experiments).

| Instrument | ACSM-UU | | ACSM-RUG | |
|---|---|---|---|---|
| Measurements | Cabauw ambient air | | chamber experiment | |
| Value | $R_{pAmN}$ | $R_{pON}$ | $R_{pAmN}$ | $R_{pON}$ |
| Upper limit | | 0.0072[a] | | 0.0035[b] |
| Geometric mean | 0.0237 ± 0.0009 | 0.0008[c] | 0.0115 ± 0.0002 | 0.0006[c] |
| Lower limit | | 0.0001[d] | | 0.0001[d] |

[a] Calculated using $RoR$ = 3.29. The $RoR$ is obtained using $R_{pON}$ from the upper limit experiment of glyoxal+$NO_3$ and measured $R_{pAmN}$, using ACSM-RUG (see Fig. 2).

[b] Experimental $NO_x^+$ ratio value from the lower limit experiment of glyoxal+$NO_3$ using ACSM-RUG (see Fig. 2a).

[c] Geometric mean of the upper and lower limit $R_{pON}$.

[d] Set as the lowest possible $NO_x^+$ ratio in CV-based instruments, approaching zero.

Since the $R_{pAmN}$ values are quite different for the two ACSMs used in this study as we see in Table 2, the upper limit $R_{pON}$ are also different by almost a factor of 2. However, since the lower limit of $R_{pON}$ approaches zero, the geometric means of the upper and lower limits for the two instruments differ by only 25%. While calibrating every instrument with a pure organic nitrate aerosol standard would be a preferable way to establish $R_{pON}$, we recognize the unlikeliness of that for all monitoring ACSMs. Therefore, we recommend this $RoR$-based approach. As will be shown in the following section, despite the uncertainties outlined here based on potential impurities in our "pure pON" chamber experiment, the nitrate splitting performs encouragingly well, with both ACSMs.

# 5 Development of $NO_x^+$ ratio method for CV-UMR-ToF-ACSM

## 5.1 Challenges in $NO_x^+$ ratio method application to CV-UMR-ToF-ACSM dataset

Applying the $NO_x^+$ ratio method to separate pAmN and pON in CV-UMR-ToF-ACSM datasets is a greater challenge than with SV, due primarily to the higher detection limit of $NO_3$. The detection limit of $pNO_3$ in CV-UMR-ToF-ACSM is 0.01-0.08 μg m$^{-3}$ (from this work and Zheng et al. (2020) converted), around 10-100 times higher compared to those in SV-cToF-AMS ($\sim$0.6 ng m$^{-3}$ from Drewnick et al. (2009)) and SV-HR-ToF-AMS (0.1-4.0 ng m$^{-3}$ from DeCarlo et al. (2006)), all converted to 10 min time resolution.

The poor detection limit for $NO_x^+$ ratios in CV-ToF-ACSM results from the low signal for $m/z$ 46 relative to $m/z$ 30 that are used to calculate frag_$NO_3$[30] ($C_{NO^+}$) and frag_$NO_3$[46] ($C_{NO_2^+}$). For instance, using the ACSM-UU, the detection limit of $NO_2^+$ is comparable to the detection limit of $NO^+$ at $pNO_3$ concentration near the detection limit of $pNO_3$ ($C_{DL,NO_2^+}$ = 0.044 μg m$^{-3}$; $C_{DL,NO^+}$ = 0.066 μg m$^{-3}$; for $C_{DL,pNO_3}$ = 0.075 μg m$^{-3}$; all in 10 min time resolution). However, the magnitude of observed $NO_2^+$ from ambient measurements is 25-500 times lower than $NO^+$ in CV-ToF-ACSM. This means the $NO_2^+$ signal intensity is regularly close to the detection limit, particularly when the total $pNO_3$ concentration is low. This behavior also leads to noisy $R_{obs}$, due to a computation of very low or negative $NO_2^+$ signals, poor baseline, or both.

## 5.2 Data pre-treatment: Time averaging and data filtering

To determine which data points are reliable for $R_{obs}$ calculation in the dataset, we could discard observed $NO_2^+$ signal intensities that are below the detection limit. However, this would result in removing nearly all the data, including data that, while low and noisy, can still provide quantitative information with adequate averaging. Therefore, we use observed $NO^+$ signals as the filtering parameter. The $NO^+$ signal accounts for $\sim$95% of the total concentration of $NO_3$ species measured by ToF-ACSM (no RIE applied) and thus is a good indicator of when both $NO^+$ and $NO_2^+$ signals are too uncertain.

Eq. 4 describes the $NO^+$ signal limit ($C_{NO^+,lim}$) which assures reliable separation of $f_{pON}$ and $f_{pAmN}$ calculated using the detection limit of $NO_2^+$ ($C_{DL,NO_2^+}$) and $R_{pAmN}$ as filter $NO_x^+$ ratio. We choose the larger $R_{pAmN}$ value, which is a less strict limit relative to $R_{pON}$ value, but still keeps any data with sufficiently good signal-to-noise ratio. The measured data points with observed $NO^+$ signal intensity below these criteria are replaced with not-a-number (nan).

$$C_{NO^+,\,lim} = \frac{C_{DL,NO_2^+}}{R_{pAmN}} \tag{4}$$

On this basis, we recommend data pre-treatments by time averaging and data filtering using observed $NO^+$ signal contribution as parameter. Time averaging over longer time periods allows the reduction of the electronic noise coming from the instrument response and low counting statistics associated with sampling ambient air. Meanwhile, the data filtering serves to determine the minimum $pNO_3$ concentration at which reliable $R_{obs}$ can be obtained to calculate $f_{pON}$ and $f_{pAmN}$.

The values of $C_{DL,NO_2^+}$ and $C_{NO^+,lim}$ in different time averaging are evaluated in Table 3 for the CV-ToF-ACSM deployed for ambient measurements in Cabauw, the Netherlands. The signal limit is lower as the time resolution increases due to the

improvement of detection limit with better statistics. For measurements in this study, the minimum reliable $C_{\mathrm{NO^+}}$ for 10 min, 30 min, 60 min, and 120 min time resolution are 1.88, 1.08, 0.77, and 0.54 µg m$^{-3}$, respectively.

With these $C_{\mathrm{NO^+,lim}}$, we performed data filtering to the ambient measurement time series from Cabauw in different averaging of the time series. The time averaging (generated by Tofware v3.3) is applied first before the data filtering to maximize retained data in the concentration average.

**Table 3.** Detection limits of $NO_2^+$ and signal limits for $NO^+$ across different time averaging with $R_{\mathrm{pAmN}}$ as filter $NO_x^+$ ratio for the CV-UMR-ToF-ACSM deployed for ambient measurements in the rural site of Cabauw, the Netherlands (ACSM-UU).

| Signal intensity (µg m$^{-3}$) | 10 min | 30 min | 60 min | 120 min |
|---|---|---|---|---|
| $C_{\mathrm{DL,NO_2^+}}$ | 0.044 | 0.026 | 0.018 | 0.013 |
| $C_{\mathrm{NO^+,lim}}$ (filter: $R_{\mathrm{pAmN}} = 0.0237$) | 1.88 | 1.08 | 0.77 | 0.54 |

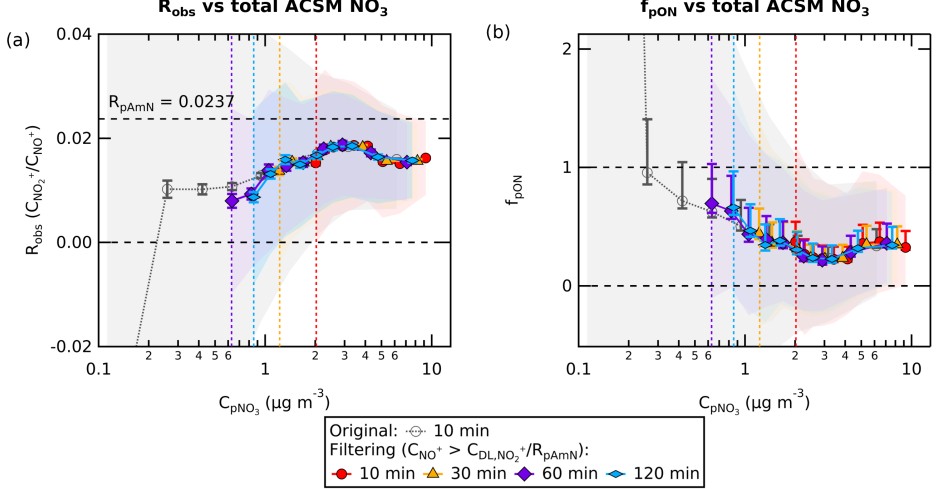

**Figure 3.** Chemical coordinate plots (a) between $R_{\mathrm{obs}}$ against $C_{\mathrm{pNO_3}}$ in Cabauw (net 205 days of data), and (b) between the $f_{\mathrm{pON}}$ calculated using geometric mean of $R_{\mathrm{pON}}$ ($R_{\mathrm{pON}} = 0.0008$) against $C_{\mathrm{pNO_3}}$. The revised fragmentation table for typical ambient dataset is used to obtain $C_{\mathrm{NO^+}}$ and $C_{\mathrm{NO_2^+}}$. The line and marker traces represent the quantile average. The colored shading represents the standard deviation of each quantile while the whisker is the standard error. The standard deviation and standard error both include the uncertainty of ion counting statistics from measurements and uncertainty from ODR fit slope of fragmentation table correction. For $f_{\mathrm{pON}}$ (plot b), the standard deviation and standard error also include the uncertainty from lower and upper limit of $R_{\mathrm{pON}}$ ($R_{\mathrm{pON}} = 0.0001$; $R_{\mathrm{pON}} = 0.0072$). All analyses were done using the 10 min, 30 min, 60 min, and 120 min averaging of the time series, with data filtering. The original 10 min data without pre-treatment is also included as comparison. The combination of data filtering and time averaging reduces the noise compared to original data and improves the minimum concentration reliable for apportionment calculation (pNO$_3$ concentration limit for each time average, as indicated with vertical dashed lines).

Chemical coordinate plots of $R_{\mathrm{obs}}$ and $f_{\mathrm{pON}}$ against the total concentration of particulate nitrate ($C_{\mathrm{pNO_3}}$) are shown in Fig. 3. These plots show the quantile average of the output variable on the y-axis (i.e. $R_{\mathrm{obs}}$ or $f_{\mathrm{pON}}$) as a function of concentration bins in the x-axis ($C_{\mathrm{pNO_3}}$). Fig. 3a,b shows that data filtering removes extreme $R_{\mathrm{obs}}$ and $f_{\mathrm{pON}}$ values near detection limit level. The fact that the chemical coordinate trends are consistent across different averaging times indicates that filtering to remove noisy data will not bias the interpretation of the ensemble dataset. The value of treating the data as a chemical coordinate plot is to allow for a robust characterization of the average trend, even using a method with substantial uncertainties. Importantly, the standard errors represent how well the averages are known. Their small uncertainty ranges support that the trend characterized is robust.

The combination of data filtering and time averaging shows different concentration cut-off for calculation of the $\mathrm{NO_x^+}$ ratio. The concentration cut-off is lower for longer averaging times due to the improvement of $C_{\mathrm{DL,NO_2^+}}$. For measurements in this study, the $C_{\mathrm{pNO_3}}$ cut-off for 10 min, 30 min, 60 min, and 120 min time averaging are 2.0, 1.2, 0.9, and 0.6 µg m$^{-3}$, respectively. Because there is a trade-off between time resolution and the concentration cut-off, for a given dataset, the timescale of typical variations should be assessed in order to determine the appropriate averaging time.

## 5.3 Propagation of uncertainty

We propagated uncertainties from the variables in $f_{\mathrm{pON}}$ using simplified propagation of uncertainties using standard error (Day et al. (2023); see Section S4.3 of SI for details). The uncertainty of the final function ($s_f$) is calculated using the standard error ($s_{x_i}$) and partial derivative of the function ($\frac{\partial f}{\partial x_i}$) of each measurand ($x_i$) using Eq. 5.

$$s_{f(x_i, x_{i+1}, \ldots)} = \sqrt{\sum_{i=1}^{N} \left( \frac{\partial f(x_i, x_{i+1}, \ldots)}{\partial x_i} \right)^2 \cdot s_{x_i}^2} \tag{5}$$

The uncertainty of $f_{\mathrm{pON}}$ arises from three terms: $R_{\mathrm{obs}}$, $R_{\mathrm{pAmN}}$, and $R_{\mathrm{pON}}$. For $R_{\mathrm{obs}}$, the uncertainty is further composed of 6 components that make up $\mathrm{NO^+}$ and $\mathrm{NO_2^+}$ signals (the uncertainties of $m/z$ 29, $m/z$ 30, $m/z$ 45, and $m/z$ 46 related to precision uncertainty from electronic noise and ion counting statistics, and the uncertainties of the fragmentation table multipliers, represented by the uncertainty of ODR fit slope). For $R_{\mathrm{pAmN}}$, we propagate the uncertainty of the mean $\mathrm{NO_x^+}$ ratio from repeated $\mathrm{NH_4NO_3}$ measurements using ODR fit to consider the instrument stability in acquiring the $\mathrm{NO_x^+}$ ratio over time. For $R_{\mathrm{pON}}$, the uncertainty is set to zero. Instead, the lower and upper limit of $R_{\mathrm{pON}}$ (see Table 2) serve to give a range of final propagated uncertainty, which includes $R_{\mathrm{obs}}$, $R_{\mathrm{pAmN}}$, and $R_{\mathrm{pON}}$.

Fig. 4a shows that low pNO$_3$ concentrations produce larger uncertainties in $f_{\mathrm{pON}}$ ($s_{f_{\mathrm{pON}}}$) compared to higher concentrations. If we compare the analysis with and without data filtering, we observe that filtering targets data points with high absolute $s_{f_{\mathrm{pON}}}$ (above $\pm 0.5$). We avoid removing many data points in the low concentration range by performing time averaging, where the average uncertainty decreases by $\sim \sqrt{N}$ for each $N$-fold of averaging from 10 min. Meanwhile, the uncertainties from $R_{\mathrm{pAmN}}$ and $R_{\mathrm{pON}}$ ($RoR$) remain unaffected by time averaging because the values remain constant in the time series.

Several studies reported $f_{\mathrm{pON}}$ lower than 20%, which occur mainly in urban areas and during a colder period. Yu et al. (2024) observed a lower range of annual average of urban $f_{\mathrm{pON}}$ in China to be $\sim 17\%$, while Mohr et al. (2012) and Pandolfi et al.

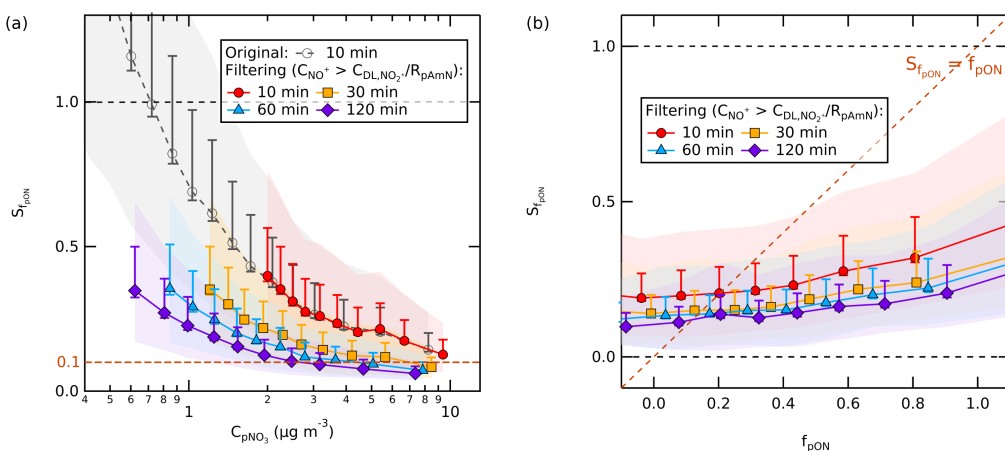

**Figure 4.** The chemical coordinate plot (quantile average) between (a) $s_{f_{pON}}$ and $C_{pNO_3}$ (logarithmic scale), and (b) $s_{f_{pON}}$ and $f_{pON}$ (linear scale), with $R_{pAmN} = 0.0237$ as filter $NO_x^+$ ratio at various averaging of the time series. The line and marker trace represents the average uncertainty produced from the geometric mean of $f_{pON}$. The uncertainty consists of uncertainties of ion counting statistics from measurements, uncertainty from ODR fit slope of fragmentation table correction, and uncertainty of $R_{pAmN}$. The colored shading represents the standard deviation of each quantile, while the whisker is the standard error. The shading and whisker both include the uncertainty of $R_{pON}$ coming from the lower and upper limit of $R_{pON}$ ($R_{pON} = 0.0001$ and $R_{pON} = 0.0072$), and also the uncertainty of the average quantile. Uncertainties of $f_{pON}$ <0.1 (absolute value) is reached at pNO3 concentration >10 μg m$^{-3}$ for 10 min time averaging, while at 60 min, it is reached already at ~4 μg m$^{-3}$. In terms of fraction, uncertainties of $f_{pON}$ below the calculated $f_{pON}$ ($s_{f_{pON}} < f_{pON}$) is reached at $f_{pON}$ ~0.2 for 10 min averaging, while at 60 min, it is reached at $f_{pON}$ ~0.17.

(2014) reported ~13% fraction in Barcelona, Spain, and Xu et al. (2021) reported 9.8% fraction in wintertime Beijing, China. If we use the lower range of $f_{pON}$ of ~ 10% as reference for the minimum uncertainty needed to report reliable $f_{pON}$, we can observe that the lowest pNO3 where we obtain below 0.1 absolute uncertainty in $f_{pON}$ decreases along with time averaging as well. Uncertainties below ±0.1 can only be reached at pNO3 concentration higher than 10, 7, 4, 2 μg m$^{-3}$ at 10, 30, 60, and 120 min time averaging, respectively.

Fig. 4b shows the relationship between the absolute $s_{f_{pON}}$ and $f_{pON}$. The limit at which the absolute value of $s_{f_{pON}}$ is below or equal to $f_{pON}$ (minimum uncertainty) is found to be 20%, 15%, 14%, 12% at 10, 30, 60, and 120 min time averaging, respectively. This result suggests that the $NO_x^+$ ratio method in CV-UMR-ToF-ACSM is more reliable to analyze nitrate pollution episodes or chamber experiments, and not for low background pNO3 concentrations. By combining both the concentration limit and the fraction limit, we suggest that in the region where pNO3 concentration is <10 μg m$^{-3}$ and/or $f_{pON}$ <12%, the

method requires a longer time average to calculate $f_{pON}$ to achieve minimum uncertainty.

# 6  Case studies demonstrating $NO_x^+$ ratio method for CV-ToF-ACSM

## 6.1  Ambient measurements at rural site

In order to demonstrate the efficacy of this method, we investigate the trend of $f_{pON}$ and $f_{pAmN}$ in the dataset observed at Cabauw. The ambient concentration time series of pAmN and pON (using $R_{pAmN} = 0.0237$ and $R_{pON} = 0.0008$) is compared with the ACSM measured Org and NH4 fractions, shown in Fig. 5. The time series is averaged to 60 min and filtered using $R_{pAmN} = 0.0237$ (values of $C_{NO^+} < C_{DL,NO_2^+}/R_{pAmN}$ are discarded). We observe that $f_{pON}$ increases with increasing fraction of organic aerosol concentration in total ACSM $PM_{2.5}$ ($C_{OA}/C_{PM_{2.5}}$, from 48% to 64%), whereas the $f_{pAmN}$ increases with increasing fraction of particulate ammonium concentration in total ACSM $PM_{2.5}$ ($C_{pNH_4}/C_{PM_{2.5}}$, from 8% to 15%). This shows that the organic nitrate fraction is correlated with availability of organics (particularly at high organics fraction), while the inorganic fraction increases with available $NH_4$ (particularly at low ammonium fraction).

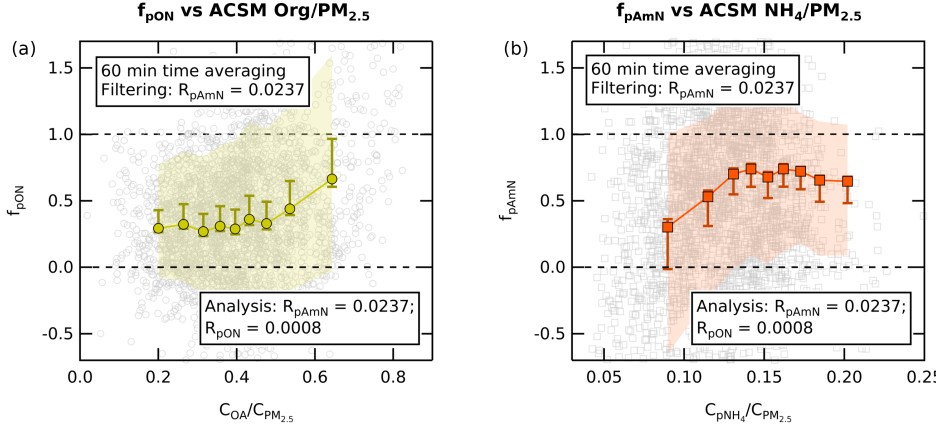

**Figure 5.** (a) The chemical coordinate plot (quantile average) of $f_{pON}$ against $C_{OA}/C_{PM_{2.5}}$ shows an average increase of $f_{pON}$ as OA fraction increases from 48% to 64%, where the $f_{pON}$ varies from 33% to 66%. (b) The chemical coordinate plot (quantile average) of $f_{pAmN}$ against $C_{pNH_4}/C_{PM_{2.5}}$ shows an average increase of $f_{pAmN}$ as pNH$_4$ fraction increases from 8% to 15%, where the $f_{pAmN}$ varies from 30% to 74%. Note: All quantile averages were calculated using the 60 min averaging of the time series. The colored shading is the standard deviation of each quantile average, while the whiskers represent the standard error. They include the uncertainties of ion counting statistics from measurements, uncertainty from ODR fit slope of fragmentation table correction, and $R_{pON}$ range value.

We also investigate specific nitrate episodes to show the composition of pON and pAmN in ambient pollution events. The time series of the $NO_x^+$ ratios, ACSM species concentrations (OA, pNH$_4$, pNO$_3$), pAmN and pON concentrations are shown in Fig. 6. Four nitrate episodes from spring (15 May 2023), summer (23 June 2023), autumn (05-06 September 2023), and winter (11 January 2024) are shown. The uncertainty of the concentration is obtained by combining the uncertainty from the nitrate fraction ($f_{pON}$ or $f_{pAmN}$), the total ACSM NO$_3$, and the $R_{pON}$ value range (see Eqs. S14 and S15). Note that the reported uncertainties are only related to precision uncertainty, and not to concentration quantification (e.g., ionization efficiency) like

the one described by Bahreini et al. (2009) (see details in Section S4.3 of SI). We observe that the adapted $NO_x^+$ ratio method is able to separate contributions of pON and pAmN to the total $pNO_3$ concentration. In Fig. 6i,ii,iv, we can see that the time series of pON tracks with total OA, while pAmN tracks with $pNH_4$ in the rural site. In Fig. 6iii, no significant trend is observed for pON due to lower mass loading of $pNO_3$.

Unless there are co-located ambient measurements of UMR and HR instruments, the reported concentration of $NO_3$ in pON depends on the value of $R_{pON}$ and multipliers used to calculate $NO^+$ and $NO_2^+$ signal contribution in the fragmentation table. The sensitivity of these variables needs to be assessed to understand which parameter is the most critical in the separation of inorganic and organic nitrate signal from ACSM.

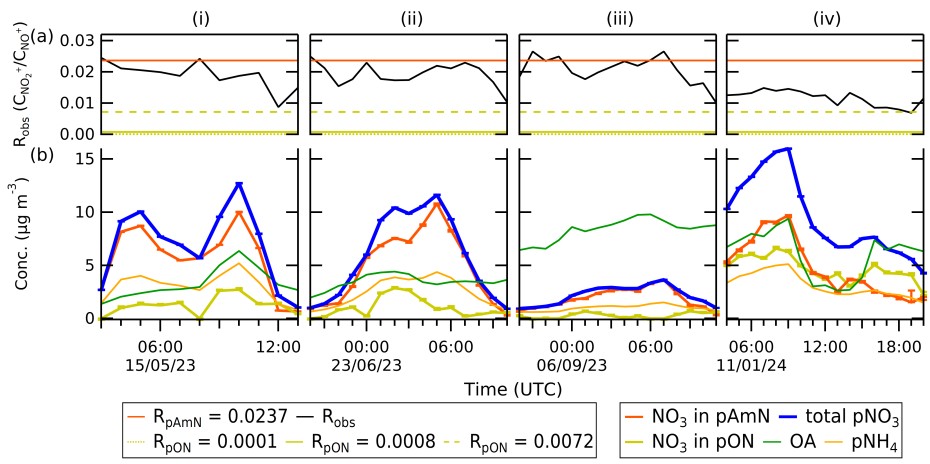

**Figure 6.** The time series for select periods of (a) $R_{obs}$, with horizontal lines indicating the values for $R_{pAmN}$ and $R_{pON}$, (b) mass concentration of ACSM-measured total OA, $pNH_4$, and $pNO_3$, and pAmN and in pON. The separation of pAmN and pON is calculated using $R_{pAmN}$ = 0.0237 and $R_{pON}$ = 0.0008. The whiskers represent the uncertainty from nitrate fraction ($f_{pON}$ or $f_{pAmN}$), precision uncertainty of total ACSM $NO_3$ from Tofware v3.3, and the $R_{pON}$ value range. The time series (i), (ii), (iii), (iv) represents representative composition in spring, summer, autumn, and winter, all in 60 min time averaging.

In Fig. 7, we show the sensitivity analysis of $R_{pON}$, $a_{Org[30],[29]}$ and $a_{Org[46],[45]}$. We varied $R_{pON}$ from zero to $R_{pON}$ = 0.0072 (calculated using $RoR$ = 3.29). Fig. 7a suggests that for $R_{pON} \leq 10^{-3}$, the reported pON concentrations are not significantly different and therefore confirm the lower limit of $R_{pON}$ approaching zero, as established in Table 2.The value of $R_{pON}$ calculated using $RoR$ (in this case $R_{pON}$ = 0.0072) shows relatively higher pON concentration, which is consistent with its use as the upper limit of $R_{pON}$.

We also varied $a_{Org[30],[29]}$ and $a_{Org[46],[45]}$ using the values listed in Table S4 and Fig. S1c,d in SI. Fig. 7b shows that the calculated pON concentration is sensitive to $a_{Org[46],[45]}$, which is not the case for $a_{Org[30],[29]}$. It further demonstrates that the limitation of this adapted $NO_x^+$ ratio method is its sensitivity towards $a_{Org[46],[45]}$ to obtain $NO_2^+$ signal contribution. Chemically, this suggests that the organic contribution in $m/z$ 46 can vary and comprises a substantial portion of the total $m/z$ 46 signal. Therefore, an average correction for $m/z$ 46 may result in a high uncertainty of calculated $f_{pON}$ using the $NO_x^+$ ratio method.

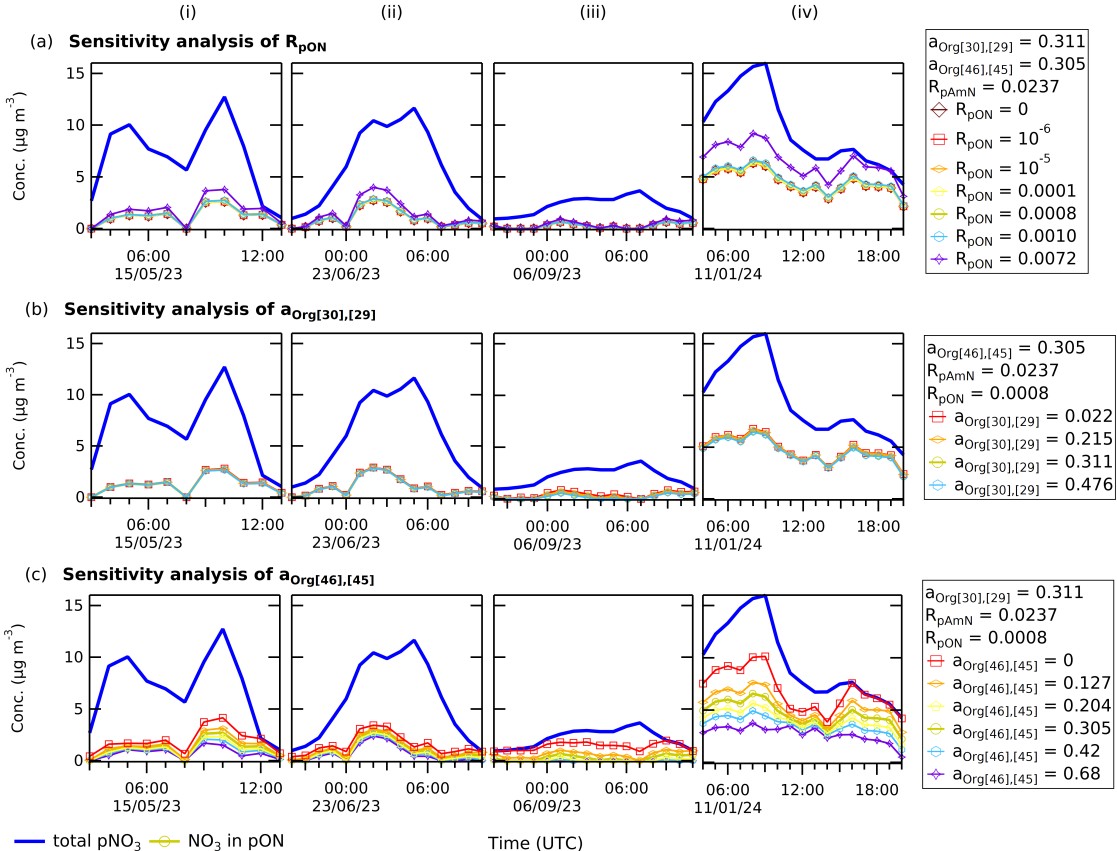

**Figure 7.** Sensitivity analysis of (a) $R_{\mathrm{pON}}$, (b) $a_{\mathrm{Org[30],[29]}}$, and (c) $a_{\mathrm{Org[46],[45]}}$ to the pON concentration ($\mathrm{NO_3}$ in pON) calculated using adapted $\mathrm{NO_x^+}$ ratio method. The time series in each case is an ambient pollution episode in Cabauw, the Netherlands, during spring, summer, autumn, and winter period (i-iv). The results show that (a) $R_{\mathrm{pON}} \leq 10^{-3}$ does not show significant differences in reported pON concentration, and (b) the reported pON concentration is not sensitive to the change of $a_{\mathrm{Org[30],[29]}}$. In contrast, the results (c) show a significant change in reported pON concentration when $a_{\mathrm{Org[46],[45]}}$ is varied, showing that this correction is the primary limitation of the $\mathrm{NO_x^+}$ ratio method in CV-ACSM, because it can be highly dependent on the calculation of $\mathrm{NO_2^+}$ signal contributions to m/z 46.

This was considered in the propagation of uncertainty, where we take into account the changing $a_{\mathrm{Org[46],[45]}}$ in the reported $f_{\mathrm{pON}}$, therefore representing a range of the observed organic nitrate contribution.

## 6.2 pAmN and pON formation in chamber experiment

We investigated the pAmN and pON formation in a chamber experiment using limonene precursor, $\mathrm{NO_3}$ oxidant, and with AmN seed aerosol. The experiment was carried out in the AIDA chamber. Alongside CV-UMR-ToF-ACSM (ACSM-RUG), a 420 SV-HR-ToF-AMS managed by IMK KIT was also deployed.

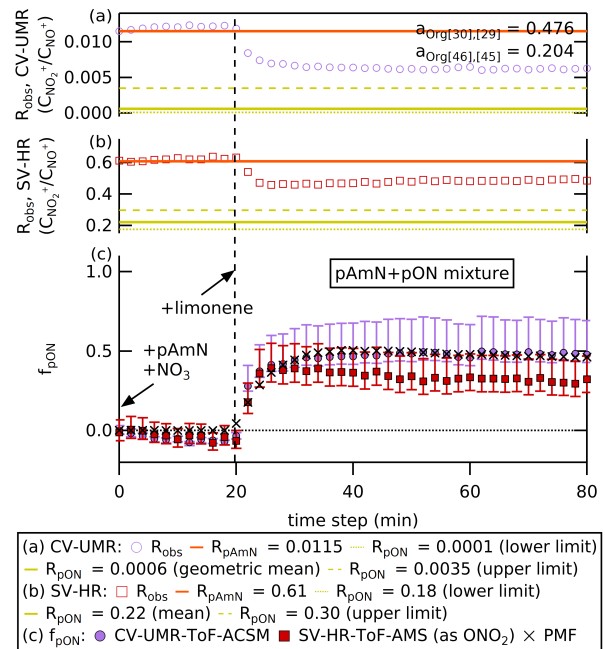

**Figure 8.** (a,b) The time series in 2 min time averaging of $R_{obs}$, $R_{pAmN}$, and $R_{pON}$ measured by CV-UMR-ToF-ACSM (top) and SV-HR-ToF-AMS (middle). The fragmentation table specific for terpene is used to obtain the $C_{NO^+}$ and $C_{NO_2^+}$ of the chamber experiment. (c) The time series in 2 min time averaging of $f_{pON}$ from $NO_x^+$ ratio method applied to SV-HR-ToF-AMS and CV-UMR-ToF-ACSM, as well as PMF method applied to CV-UMR-ToF-ACSM. The markers represent geometric mean for $NO_x^+$ ratio method applied to CV-UMR-ToF-ACSM (circle), mean for $NO_x^+$ ratio method of SV-HR-ToF-AMS (square) and for PMF method of CV-UMR-ToF-ACSM (cross). The whiskers represent the uncertainties from the value range of $R_{pON}$ combined with the uncertainties from electronic noise, ion counting statistics, fragmentation table (for UMR), and $R_{pAmN}$. The uncertainty from the PMF analysis is not shown for simplicity.

The ACSM data is analyzed using the revised fragmentation table specific for terpene chamber experiments (see Section 3.2), while the AMS data is analyzed using Squirrel 1.66E and PIKA 1.26E with the default ion list. Although the time resolution is 2 min, time averaging and data filtering is not necessary since the experiment involves high concentrations. The lower limit, geometric mean, and upper limit of $R_{pON}$ ($R_{pON}$ = 0.0001; 0.0006; 0.0035, respectively) are employed to estimate the
uncertainty of $f_{pON}$, alongside $R_{pAmN}$ = 0.0115 (see Table 2). For the AMS instrument, the measurements of pure pAmN give $R_{pAmN} = 0.61 \pm 0.05$ and the $R_{pON}$ value is calculated using $RoR = 2.75 \pm 0.70$ (Day et al., 2022a), which gives $R_{pON}$ = 0.18; 0.22; 0.30 as lower limit, mean, and upper limit, respectively.

We also performed PMF analysis using the ACSM data including OA, $NO_x^+$, and $NH_x^+$ ions, which has been similarly done in other studies (e.g., Day et al. (2022a) and references therein). The 2 min average matrices of UMR organic fragment mass
spectra with a $m/z$ of 12 to 120, fragments of ammonium ($NH_4\_16$ and $NH_4\_17$, two main signals of $NH_4$ which are $NH_2^+$ and $NH_3^+$), and fragments of nitrate ($NO_3\_30$ and $NO_3\_46$, two main signals of $NO_3$ which are $NO^+$, $NO_2^+$) are used as

variables in the PMF input matrix. Fragment contributions are calculated using the terpene-related fragmentation table (see Section 3.2). We choose a two-factor solution (see Fig. S2) because we are interested in splitting the aerosol mass only into inorganic aerosol (pAmN) and organic aerosol (OA mixture, containing pON). From the organic aerosol factor, we calculate

the $f_{pON}$ from the factor concentration time series. The details of the PMF method are described in Section S5, including the statistical summary and the diagnostic plots of the PMF analysis.

The $NO_x^+$ ratio and $f_{pON}$ time series for the limonene SOA experiment are shown in Fig. 8. The whiskers around the mean $f_{pON}$ represents the uncertainties coming from the value range of $R_{pON}$ combined with the uncertainties from ion counting statistics, electronic noise, fragmentation table (for UMR), and $R_{pAmN}$ (see Section 5.3). The initial mixture of $NH_4NO_3$ seed

and $NO_3$ radical from $NO_2$ and $O_3$ gives a ratio that matches $R_{pAmN}$ from an offline calibration, where $f_{pON}$ is close to zero for both instruments. Although nitric acid ($HNO_3$) can be formed by nitrogen pentoxide ($N_2O_5$) hydrolysis under these humid conditions, experiments were run in excess $NH_3$ and therefore we expect no substantial increase of $HNO_3$ that can affect $NO_x^+$ ratio. After the limonene injection, the $R_{obs}$ rose to values in between $R_{pAmN}$ and $R_{pON}$, indicating the formation of a mixture of pAmN and pON inside the chamber.

Based on observations of Takeuchi et al. (2024), pON measured with the SV-AMS is only quantitatively detected as $-NO_2$ moiety, not $-ONO_2$. The reasoning for this difference in detection of alkyl nitrate ($RONO_2$) and $NH_4NO_3$ is thought to be due to thermal decomposition of pON producing $NO_2$ gas in the vaporizer, while $NH_4NO_3$ would more likely decompose to $HNO_3$, and thus dominantly ionizing gases of different molecular weights. The "missing oxygen" is generally retained bound to the carbon, and thus accounted as organic moiety. Thus, $f_{pON}$ needs to be corrected for this phenomenon using molar mass

ratio of $NO_3/NO_2$ (62 g mol$^{-1}$/46 g mol$^{-1}$) to recalculate the pON mass. There is no study yet assessing the necessity of correcting $f_{pON}$ in CV, and we suggest that it is likely unnecessary due to the more complete thermal decomposition that shifts the fragmentation pattern of both $NH_4NO_3$ and $RONO_2$ to $NO_2$ and $NO$.

Both $NO_x^+$ ratio methods (SV-AMS and CV-ACSM) and the PMF method (CV-ACSM) show a similar response to the injection of limonene, whereupon $f_{pON}$ increases rapidly from ~0 to 0.3-0.5. The agreement between the two instruments on

this initial pON production is encouraging. However, after this initial jump, the trend of $f_{pON}$ seems to vary. While mean $f_{pON}$ calculated using $NO_x^+$ ratio method from CV-ACSM remains steady after the injection, the other two results show a gradual decrease of $f_{pON}$ as the chamber dilutes.

The $NO_x^+$ ratio method on CV-ACSM data shows a similar change in fpON relative to the PMF method immediately after limonene injection but then continues to decrease over time (see Fig. 8). The PMF method combines the variations of Org,

$NO_x^+$, and $NH_x^+$ ions to obtain the factor profiles, therefore allowing a more subtle change in the chamber composition to be taken account. The $NO_x^+$ ratio method in CV-UMR, in contrast, only takes into account fragments that are in $m/z$ 30 and $m/z$ 46 using constant fragmentation table relationship.

Similarly, the SV-AMS also shows a gradual decrease of mean $f_{pON}$ unlike the $NO_x^+$ ratio method from CV-ACSM (see Fig. 8). This suggests that changing contributions of organics at $m/z$ 30 and $m/z$ 46 may be taken into account by HR peak fitting

and not by the UMR fragmentation table, causing a divergence as the chamber aerosol dilutes. Based on the sensitivity analysis in Fig. 7, the signal contribution of $NO_2^+$ ($m/z$ 46) is the largest source of uncertainty in the CV-ACSM, since the adapted

method is sensitive to the change of $a_{\text{Org}[46],[45]}$. On the other hand, the uncertainty of $RoR$ used to calculate $R_{\text{pON}}$ accounts for the largest contribution to the uncertainty calculated for SV-AMS. Since the $RoR$ in Day et al. (2022a) relies on the average value of a broad range of organic nitrate, a chamber experiment that uses a specific precursor is likely to have $R_{\text{pON}}$ further away from the average value than, for instance, a complex ambient mixture.

Nevertheless, we are encouraged by the match in responses upon formation of organic nitrate, indicating that the $NO_x^+$ ratio method is similarly sensitive to changing nitrate speciation in both instruments. When considering the propagation of uncertainty, we observe overlaps between the results of CV-ACSM and SV-AMS. Therefore, by considering the uncertainty from $a_{\text{Org}[46],[45]}$ in CV-UMR and uncertainty from $RoR$ in SV-AMS, the results for both instruments are comparable. Further investigation of the detailed response of each instrument to changing aerosol composition would be valuable.

## 7   Conclusions and recommendations

We have shown the separation of particulate ammonium nitrate (pAmN) and particulate organic nitrate (pON) signal from total particulate nitrate (pNO$_3$) signal measured using time-of-flight aerosol chemical speciation monitor equipped with capture vaporizer in unit mass resolution (CV-UMR-ToF-ACSM), using an adapted $NO_x^+$ ratio method with a revised fragmentation table and data pre-treatment. The shift of fragmentation pattern towards smaller ion fragments in the capture vaporizer (CV) compared to the standard vaporizer (SV) affects the signals of $NO^+$ and $NO_2^+$ fragments and interferences by Org fragments used to calculate the $NO_x^+$ ratio in UMR. Therefore, we recommend updating the default fragmentation table from Allan et al. (2004) for entries shown in Table 4 before applying the $NO_x^+$ ratio method, according to the aerosol composition. As noted previously, substantial corrections to the fragmentation table for these terms have been shown to be needed for measurements using the SV under some conditions (Fry et al., 2018).

**Table 4.** Proposed $m/z$ 30 and 46 entries for Org and NO$_3$ in the revised fragmentation table adapted for $NO_x^+$ ratio method in CV-UMR-ToF-ACSM. The multipliers ($a_{\text{Org}[x],[i]}$) are applied according to the aerosol composition. The entries that are not included in this table should follow the fragmentation table of Allan et al. (2004) for AMS and adapted without gas-phase corrections (frag_air) for ACSM.

| $m/z$ | Revised fragmentation table for CV-UMR-ToF-ACSM | | $a_{\text{Org}[x],[i]}$ | | | |
| | Org | NO$_3$ | General | Biogenic[a] | Glyoxal | Terpene |
|---|---|---|---|---|---|---|
| 30 | $a_{\text{Org}[30],[29]}$*frag_Org[29] | [30], -frag_Org[30] | 0.311 | 0.32 | 0.291 | 0.476 |
| 46 | $a_{\text{Org}[46],[45]}$*frag_Org[45] | [46], -frag_Org[46] | 0.305 | 0.68 | 0.082 | 0.204 |

[a]Retrieved from Hu et al. (2017).

The shift of fragmentation pattern in the CV towards more formation of $NO^+$ fragments and less of $NO_2^+$ fragments changes the magnitude of the $NO_x^+$ ratio for both pure pAmN and pure pON. The $NO_x^+$ ratio in CV is affected by the aerodynamic lens alignment, and therefore we recommend users to align their aerodynamic lens to obtain the correct $NO_x^+$ ratio.

To separate the pAmN and pON signal from total pNO$_3$ and calculate the particulate organic nitrate fraction ($f_{\text{pON}}$), the regular ammonium nitrate calibration should be used to obtain the $NO_x^+$ ratio for pure ammonium nitrate ($R_{\text{pAmN}}$), where a

value of 0.01-0.07 is expected. On the other hand, we observed from a chamber experiment that the $R_{pON}$ value approaches zero in the CV. Therefore, we recommend analyzing using three $R_{pON}$ values, to describe the upper limit ($R_{pON}$ calculated using $RoR = 3.29$), geometric mean, and lower limit ($R_{pON} = 0.0001$), which also provides $f_{pON}$ uncertainty. If possible, we recommend constraining the $RoR$ for more accurate results (e.g., performing chamber experiment to form pure organic nitrate). Through this study, we hope to inspire more research regarding $R_{pON}$ measurement in CV-based instruments to obtain more precision in analyzing organic nitrate concentrations.

The observed $NO_x^+$ ratio ($R_{obs}$) tends to have more noise in CV-based measurements compared to SV. Data filtering using the instrument's $R_{pAmN}$ and $NO_2^+$ detection limit has shown that the adapted $NO_x^+$ ratio method in the CV-UMR-ToF-ACSM is able to filter unreliable measurements with concentration cut-off ranging from 0.6 to 2.0 $\mu g\ m^{-3}$, depending on time averaging. This data pre-treatment filters data points with high fraction uncertainty (above $\pm 0.5$) and decreases the average uncertainty by $\sqrt{N}$ for each $N$-fold of averaging from 10 min.

With a longer time averaging, the concentration limit and fraction limit improve, which allows more reliable determination of $f_{pON}$ and $f_{pAmN}$. The method reports absolute uncertainty of particulate organic nitrate <10% at the total particulate nitrate concentrations of 2 $\mu g\ m^{-3}$ (120 min time averaging) to 10 $\mu g\ m^{-3}$ (10 min time averaging) and organic nitrate fraction of 10% (120 min time averaging) to 20% (10 min time averaging). We recommend users to average the time series to 30 min or 60 min to retain information about real ambient variation, while improving the reliable nitrate concentration limit. This may also be convenient when comparing to auxiliary data that are typically reported half-hourly or hourly. In the region where pNO$_3$ concentration is <10 $\mu g\ m^{-3}$ and/or $f_{pON}$ <12%, longer time averaging may be necessary to achieve the absolute uncertainty <10%. In studies where noise is not a problem (e.g., chamber experiments with high particle concentration), time averaging may be unnecessary. With a similar approach, this method could be used for SV-UMR-ToF-ACSM observations as well, with a fragmentation table suited for SV-based measurements.

The adapted $NO_x^+$ ratio method on rural nitrate episodes can distinguish periods with pAmN or pON as the major component, confirmed by relation to the ammonium and organic aerosol composition, respectively. The adapted $NO_x^+$ ratio method applied to CV-UMR-ToF-ACSM measurements in a chamber experiment is able to replicate the response to precursor injection observed from PMF analysis of the same measurements, as well as a co-located high-resolution time-of-flight aerosol mass spectrometer equipped with standard vaporizer (SV-HR-ToF-AMS). The largest uncertainties in this comparison come from $a_{Org[46],[45]}$ (CV-UMR-ToF-ACSM) and $RoR$ (SV-HR-ToF-AMS). The adapted $NO_x^+$ ratio method for CV-UMR-ToF-ACSM demonstrated in this study can be used at monitoring sites to monitor regional $f_{pON}$ and improve understanding of particulate nitrate sources and evolution.

**Appendix A: List of terms and abbreviations**

Table A1: List of important terms and abbreviations used in the manuscript.

| Terms | Name |
| --- | --- |
| ACSM | Aerosol Chemical Speciation Monitor |
| ACTRIS | Aerosol, Clouds and Trace Gases Research Infrastructure |
| AIDA | Aerosol Interaction and Dynamics in the Atmosphere |
| AmN | ammonium nitrate |
| AMS | Aerosol Mass Spectrometer |
| AmS | ammonium sulfate |
| $a_{\mathrm{Org}[x],[i]}$ | multiplier for calculating frag_Org[$x$] based on the relationship between frag_Org[$x$] and frag_Org[$i$] |
| $C_\nu$ | concentration/signal of species $\nu$ |
| $C_{\mathrm{DL,NO^+}}$ | detection limit of $NO^+$ |
| $C_{\mathrm{DL,NO_2^+}}$ | detection limit of $NO_2^+$ |
| $C_{\mathrm{DL,pNO_3}}$ | detection limit of pNO$_3$ |
| $C_{\mathrm{NO^+}}$ | signal of $NO^+$ |
| $C_{\mathrm{NO^+,lim}}$ | signal limit of $NO^+$ for reliable organic nitrate and ammonium nitrate separation |
| $C_{\mathrm{NO_2^+}}$ | signal of $NO_2^+$ |
| $C_{\mathrm{OA}}$ | concentration of total organic aerosol |
| $C_{\mathrm{PM_{2.5}}}$ | concentration of total PM$_{2.5}$ |
| $C_{\mathrm{pNH_4}}$ | concentration of total particulate ammonium |
| $C_{\mathrm{pNO_3}}$ | concentration of total particulate nitrate |
| CAINA | Cloud-Aerosol Interactions in a Nitrogen-dominated Atmosphere |
| CE | collection efficiency |
| Chl | chloride species in AMS/ACSM |
| cToF | compact time-of-flight |
| CV | capture vaporizer |
| $f_{\mathrm{pAmN}}$ | fraction of particulate ammonium nitrate to the total nitrate |
| $f_{\mathrm{pON}}$ | fraction of particulate organic nitrate to the total nitrate |
| frag_NO$_3$[$x$] | total nitrate fragments in nominal $m/z$ $x$ |
| frag_Org[$x$] | total organic fragments in nominal $m/z$ $x$ |
| HR | high resolution |
| IE | ionization efficiency |
| IMK | Institute for Meteorology and Climate Research |
| IPL | intermediate pressure lens |

*Continues next page*

| KIT | Karlsruhe Institute of Technology |
| KNMI | Royal Netherlands Meteorological Institute |
| $m/z$ | mass-to-charge ratio |
| $n$ | number of data |
| $NO_x^+$ ratio | $NO_2^+$-to-$NO^+$ signal ratio |
| obs | observed data (ambient or chamber) |
| OA | organic aerosol |
| ODR | orthogonal distance regression |
| ON | organic nitrate |
| Org | total organic aerosol in AMS/ACSM |
| pAmN | particulate ammonium nitrate |
| PIKA | Peak Integration by Key Analysis |
| $PM_{2.5}$ | particulate matter with size <2.5um |
| PMF | Positive Matrix Factorization |
| $pNH_4$ | particulate ammonium |
| $pNO_3$ | particulate nitrate |
| pON | particulate organic nitrate |
| Q-ACSM | quadrupole-ACSM |
| $R_\nu$ | $NO_x^+$ ratio of $\nu$ |
| $R_{obs}$ | $NO_x^+$ ratio of observed data |
| $R_{pAmN}$ | $NO_x^+$ ratio of particulate ammonium nitrate |
| $R_{pON}$ | $NO_x^+$ ratio of particulate organic nitrate |
| $r^2$ | coefficient of determination |
| RIE | relative ionization efficiency |
| $RONO_2$ | alkyl nitrate |
| $RoR$ | ratio-of-ratios |
| RUG | University of Groningen |
| $s_f$ | uncertainty of function $f$ |
| $s_{f_{pON}}$ | uncertainty of particulate organic nitrate fraction to the total nitrate |
| $s_{x_i}$ | standard error |
| SI | Supplementary Information |
| SOA | secondary organic aerosol |
| Squirrel | Sequential Igor Data Retrieval |

| | |
|---|---|
| SV | standard vaporizer |
| ToF | time-of-flight |
| UMR | unit mass resolution |
| UU | Utrecht University |
| VOCs | volatile organic compounds |
| WRF-Chem | Weather Research and Forecasting model coupled with Chemistry |
| $x_i$ | measurand or measured value |
| $\dfrac{\delta f}{\delta x_i}$ | partial derivative of the function $f$ |

*Data availability.* The CV-HR-ToF spectra used to build the revised fragmentation table are retrieved from the open access AMS spectral database (http://cires1.colorado.edu/jimenez-group/AMSsd_CV, last access: 6 November 2024). The CV-UMR-ToF-ACSM dataset were collected as part of the Ruisdael Observatory network monitoring (https://ruisdael-observatory.nl, last access: 6 November 2024) and are available upon request. The chamber experiment measurements using SV-HR-ToF-AMS and CV-UMR-ToF-ACSM (ACSM-RUG) were collected
as part of the Cloud-Aerosol Interactions in a Nitrogen-dominated Atmosphere (CAINA) project (https://sites.google.com/view/cainaproject/, last access: 6 November 2024) in the Aerosol Interaction and Dynamics in the Atmosphere (AIDA) chamber managed by the Institute of Meteorology and Climate Research (IMK) in Karlsruhe Institute of Technology (KIT), Germany.

*Author contributions.* F.R.N.: conceptualization, investigation, writing, and editing. J.L.F. and D.A.D.: conceptualization, editing, and reviewing. R.M., R.H., S.H., J.F., J.M., and U.D.: resources, reviewing, and editing.

*Competing interests.* The contact author has declared that none of the authors has any competing interests.

*Acknowledgements.* This work has been accomplished by using data generated in the Ruisdael Observatory as part of continuous monitoring and the Cloud-Aerosol Interactions in a Nitrogen-dominated Atmosphere (CAINA) project. The Ruisdael Observatory is a scientific infrastructure co-financed by the Dutch Research Council (NWO; grant number 184.034.015). The CAINA project is a research consortium supported by NWO (grant number OCENW.XL21.XL21.112). D.A.D. was supported by NASA grant 80NSSC21K1451 and NSF
grant AGS-2131914. The authors acknowledge valuable discussions and technical help with Anandi Williams, Phil Croteau, Donna Sueper, Weiwei Hu, Yanxia Li, Jean-Eudes Petit, Olivier Favez, Evelyn Freney, Hasna Chebaicheb, and Laurent Meunier.

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
