# Peer review of "Development and validation of a $NO_x^+$ ratio method for the quantitative separation of inorganic and organic nitrate aerosol using CV-UMR-ToF-ACSM"

_Atmospheric Measurement Techniques, 2024_

## Referee Comment (RC2)

Comments on" Development and validation of a $NO_x^+$ ratio method for the quantitative separation of inorganic and organic nitrate aerosol using CV-UMR-ToF-ACSM" by Nursanto et al.

The manuscript presents a detailed analysis of the NOx+ ratio method for apportioning organic and inorganic nitrate aerosols using a capture vaporizer (CV) in the Aerosol Chemical Speciation Monitor (ACSM). The study involves three main components: revising the fragmentation table to enhance nitrate detection, evaluating the NO2/NO ratio for source apportionment, and applying this method to both real-world and chamber data. With more and more CV-ACSM being used, the clarification on the organic/inorganic nitrate will improve the understanding of this topic. In general, the manuscript is well-written and organized. After reading through the whole manuscript, I outline several concerns, along with suggestions for improvement.

1) The revised table's performance should be tested in various settings to ensure its broad applicability. The authors obtained the slope for the fragmentation table to correct nitrate at m/z 30 and 46. However, the author never showed the verification of this method with real ambient data. E.g. Hu et al. 2017 demonstrated a better agreement on the total nitrate between SV and CV after UMR fragmentation revision in a biogenic-dominated area. How about this revision in the urban or rural areas, as well as the chamber studies?

2) Are there more RoR ratios reported in the literature literature? In SV, Doug et al. (2022) checked a variety of literature to determine the final RoR ratio.

3) Potential nitric acid formation under high humidity should be addressed to ensure accurate results. Line 250, the authors conducted the chamber study at RH around 90% with NO3 radical. there will be a formation of nitric acid. Note that even in the absence of NH3, nitrate acid can still be formed under high RH with N2O5. How did the authors exclude the inferences of nitric acid to the NO2+/NO+ ratio determined by organic nitrate in this study?

4) The applicability of the NOx ratio method at specific concentration levels should include considerations of organic nitrate fractions and averaging times. In section 5.2, the authors declared the NOx ratio method can be used for total nitrate concentration at 0.6 ug m-3 and above, which is misleading. The sole detection limit of NO+ cannot determine the limited usage of this method. I think Figure 4 gives a more comprehensive overview of this method. The usage of the NOx ratio method was also limited by the fraction of pON in total nitrate and averaging time; this should be mentioned clearly in the main text main text , as well as shall be revised in the abstract and conclusion.  For, In the urban area, where the fON accounts for 10% or even less, the CV-AMS at 10 min resolution cannot be used.

5) Have the authors tried to determine the detection limits of NO3 using the NOx ratio using the method below, which is obtained in Fig. 2 in Hu et al. (2017)? Their work is for HR nitrate. Will this method lead to similar detection limits of total UMR nitrate with what was obtained in this study?

[Figure]

6) Line 350, I do not understand why authors only point out a fraction of 17%. In most of the urban areas, the pON fraction in total nitrate is less than 15%.

7) There are too many acronyms. A summary of the abbreviations and their corresponding full names in an appendix table improve manuscript accessibility.

Hu, W., Campuzano-Jost, P., Day, D. A., Croteau, P., Canagaratna, M. R., Jayne, J. T., Worsnop, D. R., and Jimenez, J. L.: Evaluation of the new capture vaporizer for aerosol mass spectrometers (AMS) through field studies of inorganic species, Aerosol Sci Tech, 51, 735-754, 10.1080/02786826.2017.1296104, 2017.

---

## Author Comment (AC1)

**Response to reviewers for "Development and validation of a NO$_x^+$ ratio method for the quantitative separation of inorganic and organic nitrate aerosol using CV-UMR-ToF-ACSM" by Nursanto, Farhan R.; Day, Douglas A.; Meinen, Roy; Holzinger, Rupert; Saathoff, Harald; Fu, Jinglan; Mulder, Jan; Dusek, Ulrike; and Fry, Juliane L.**

**(Manuscript ID: AMT-2024-191)**

We appreciate the anonymous referees for their detailed and constructive feedback on our manuscript. The valuable suggestions have significantly improved this revised version. To guide the review process, we have copied the reviewer comments in black text, renumbered for each reviewer to facilitate cross-referencing. Our responses are provided in regular blue text, and we have responded to all the referee's comments and made alterations to our paper **in bold text**. In black text, line/figure/table number refers to the number in the submitted preprint, while in blue text, line number refers to the line in the updated version of the manuscript.
* * *
**Response to RC1 (Anonymous Referee #1):** https://doi.org/10.5194/amt-2024-191-RC1

Nursanto et al. present methodological development to separate ammonium nitrate and organic nitrate signal in the time-of-flight aerosol chemical speciation monitor (TOF-ACSM) with capture vaporizer (CV). This has been a challenge due to the unit mass resolution (UMR) of the instrument, limiting the ability to separate different ions at the same nominal m/z as the NO (30) and NO$_2$ (46) signal. A further challenge is the inclusion of a CV, which is used to improve quantification of aerosol concentration but also induces more thermal fragmentation of the ions, leading to most of the nitrate signal occurring at m/z 30 (NO), limiting the ability to use the methods previously published about using the calibrated NO to NO$_2$ ratio from ammonium nitrate and the derived average ratio of pure organic nitrate aerosol, a.k.a. the "ratio-of-ratio" (RoR) method.

Using data previously collected from different aerosol mass spectrometers (AMS) and ACSMs with CV, the authors first investigated improving the fragmentation table, a tool used to separate ions at the same nominal m/z to differentiate the signal. As discussed in prior publications, a revised fragmentation table was necessary for the CV TOF-ACSM that they apply for the paper and recommend for future users. Next, they investigate the limits of quantification of the CV TOF-ACSM due to the low signal of NO$_2$, and what nominal RoR to utilize for the TOF-ACSM (which is different than what is used for an AMS with standard vaporizer). After determining the limits of quantification and error propagation, the authors provide initial results from measurements conducted at a long-term monitoring site and from a chamber experiment.

This paper is of use for the TOF-ACSM community, as there are many TOF-ACSM with CV collecting long-term measurements. As emissions change (and thus aerosol chemistry), being able to differentiate ammonium nitrate from organic nitrates is of great value, as these two different NO$_x$ reservoirs have different properties for the aerosol and provide insight into the chemistry controlling the pollution. After the authors address the following comments, the paper fits into AMT.

**RC1-1**. There is concern about frag_org[46] vs frag_org[45], as the $R^2$ is very weak. What is the general fractional contribution of frag_org[46] and frag_org[45] to the total signal (e.g., does it need to be corrected if this signal is low, especially in regards to $NO_2$)? Further, the correction of frag_org[46], as the authors conduct throughout the paper, is dependent on the aerosol being observed. As ambient aerosol is difficult to a priori know what is the origin, how much further uncertainty is introduced into this correction. E.g., looking at Figure 1d, for less oxidized organic aerosol (LO-OOA) and aerosol influenced by isoprene and a-pinene, which would all be scenarios expected to generally have high contribution of signal towards organic nitrate aerosol instead of ammonium nitrate, it appears the correction over corrects the signal at m/z 46. Wouldn't this then lead to a too low contribution of signal to $NO_2$ and thus under reporting organic nitrates?

The weak $r^2$ is due to the fact that we are including two extremes that are visible in Figure 1b,d (L166 → L206) in the manuscript: MO-OOA and LO-OOA. This is intended to include a wide range of aerosol profiles. We also would like to point out that the $r^2$ here refers to the correlation of the ODR fit to obtain the multiplier (Figure 1a,b), not when comparing the predicted/calculated Org contribution vs measured contribution (Figure 1c,d). To clarify this, we removed the $r^2$ values from Figure 1c,d (L166 → L206) in the updated manuscript and also updated the Table S4 in the Supplementary Information by moving the columns of $r^2$ and $χ^2$ next to the multiplier values. Furthermore, we also found and corrected a mistake in Table S4; the multiplier $a_{Org[30],[29]}$ from Hu et al. (2017) for biogenic mixture in CV-ToF should be 0.32, not 0.31. The final change is as follows:

**Table S4.** The multipliers $a_{Org[x],[i]}$ · frag_Org[i] for frag_Org[30] and frag_Org[46] from this study and other studies, re-applied to the simulated UMR spectra from the HR and UMR datasets.

| Work | $a_{Org[30],[29]}$ | $r^2$ (a) | $χ^2$ (a) | Predicted/ measured[b] | $a_{Org[46],[45]}$ | $r^2$ (a) | $χ^2$ (a) | Predicted/ measured[b] |
|---|---|---|---|---|---|---|---|---|
| | frag_Org[30] = $a_{Org[30],[29]}$ · frag_Org[29] | | | | frag_Org[46] = $a_{Org[46],[45]}$ · frag_Org[45] | | | |
| **this study (CV-ToF)[c]** | **0.311±0.016** | 0.88 | 8.48E-4 | **(96.9±4.9)%** | **0.305±0.037** | 0.43 | 6.79E-6 | **(82.5±9.5)%** |
| Allan et al. (2004) (default) | 0.022 | 0.88 | 8.03E-6 | (6.6±3.4)% | - | - | - | - |
| Fry et al. (2018) (SV-ToF) | 0.215 | 0.88 | 5.40E-4 | (66.0±3.3)% | 0.127 | 0.43 | 1.71E-6 | (30.3±3.7)% |
| Hu et al. (2017) (biogenic, SV-ToF) | 0.31 | 0.88 | 8.44E-4 | (96.4±4.8)% | 0.42 | 0.43 | 9.34E-6 | (121.7±14.0)% |
| Hu et al. (2017) (biogenic, CV-ToF) | 0.32 | 0.88 | 8.71E-4 | (99.7±5.0)% | 0.68 | 0.43 | 1.22E-5 | (214.4±25.5)% |

[a] the coefficient of determination ($r^2$) and chi-squared ($χ^2$) values refer to the slope of ODR fit used to obtain the values of $a_{Org[x],[i]}$.
[b] data from CV-ToF-AMS spectral database and experiments in AIDA chamber described in Table S1 and S2, fit for typical ambient dataset. Values printed in **bold** represent the best correlation for frag_Org[x]. The value in percentage is reported with $±2σ$.
[c] predicted UMR frag_Org[x] (calculated from dataset spectra, frag_Org[x] = $a_{Org[x],[i]}$*frag_Org[i]) vs measured frag_Org[x] (sum of all Org fragments in the nominal m/z $x$ of the original dataset spectra).

To demonstrate why MO-OOA and LO-OOA affect the $r^2$, we compare Figure R1a (originally from Figure 1b in the preprint) where we include MO-OOA and LO-OOA spectra in the ODR fit, with Figure R1b where we remove MO-OOA and LO-OOA. We found that when we remove MO-OOA and LO-OOA, the correlation improves ($r^2 = 0.64$ instead of $r^2 = 0.43$), while the slopes are not significantly different (0.297 instead of 0.305, respectively). However, when we compare the predicted vs measured frag_Org[46] in plot Figure R1c and Figure R1d, we can see that the correlation that included MO-OOA and LO-OOA still provides a closer result between the Org_46 from the frag table vs the measured Org_46 (82.5% instead of 79.7%), despite having worse $r^2$. Therefore, it is still preferable to use the multiplier $a_{Org[46],[45]} = 0.305$ for the typical ambient dataset.

[Figure]

Figure R1. The left-hand panels show the best ODR fits (set to zero intercept) which are found in the relationship between the signal contributions of frag_Org[46] vs frag_Org[45] when LO-OOA and MO-OOA are (a) included and (b) excluded. The right-hand panels show the predicted organic contributions (based on the multipliers obtained on the left-hand panels) at each m/z versus the measured amount. The plots show the predicted UMR frag_Org[46] against the measured total Org fragments in m/z 46. They demonstrate that the predicted frag_Org[46] for correlation excluding LO-OOA and MO-OOA underestimates more the measured frag_Org[46] (slope = 0.797) compared to the one that includes them (slope = 0.825, originally in the manuscript).

From the HR spectra in the database, frag_Org[46] and frag_Org[45] are relatively small compared to the total signal (up to 0.006) compared to Org_30 (up to 0.08), but comparable to frag_NO$_3$[46] (up to 0.001). Since the signals for frag_Org[46], frag_Org[45], and frag_NO$_3$[46] are comparable, the correction would matter. The examples are CV-HR mass spectra from three chamber experiments from Hu et al. (2018a) in CV-AMS spectral database ([http://cires1.colorado.edu/jimenez-group/AMSsd_CV/](http://cires1.colorado.edu/jimenez-group/AMSsd_CV/), last access: 13 March 2025), shown in Figure R2 below. Furthermore, Fry et al. (2018) and Hu et al. (2017) also found that CV-instruments performed this frag_NO3[46] correction well with frag_Org[45] (see Table S4 in the Supplementary information).

[Figure]

Figure R2. HR mass spectra from chamber experiments at CU chambers at Colorado University campus in Boulder, CO, US showing different precursors and chemistry: (a) α-pinene + $O_3$ in dark condition forming non-nitrate organics, (b) α-pinene + $NO_3$ + $NH_4(SO_4)_2$ seed forming nitrate-functionalized organics, and (c) δ-carene + $NO_3$ forming nitrate-functionalized organics.

In the case of scenarios where we expect higher organic nitrate than ammonium nitrate aerosol, we expect lower ambient $NO_x^+$ ratios ($R_{obs} = C_{NO2+}/C_{NO+}$), closer to the $NO_x^+$ ratio of pure organic nitrate ($R_{pON}$). Thus, if $C_{NO+}$ is constant, $C_{NO2+}$ would move towards lower values, meaning that there would be larger frag_Org[46] to subtract from total m/z 46 in order to obtain lower frag_NO3[46]. This is the case in Figure 1d for LO-OOA and aerosol produced from isoprene and α-pinene precursors when we are using the multiplier $a_{Org[46],[45]} = 0.305$, where the calculated frag_Org[46] is higher than is detected. Therefore, overcorrection would not underestimate organic nitrates but rather overestimate. To avoid this overestimation, we provided the composition-specific fragmentation table (Section 3.2, summarized in Table 4) where, for instance, aerosol formed from terpene precursors (isoprene and α-pinene) have a smaller $a$ value used to obtain frag_Org[46] ($a_{Org[46],[45]} = 0.204$), meaning a smaller correction.

However, as the referee mentioned above, we do not actually know a priori the composition of the aerosol precursors, and therefore users still have to rely on the "general" fragmentation table correction for typical ambient data set. We assume that by including HR spectra from various origins (chamber experiment, ambient measurement, and lab standards) to build the "general" fragmentation table, the uncertainty coming from the ODR fit already includes the uncertainty from the range of potential ambient mixtures. It has been considered in the propagation of uncertainties, under the same terms as uncertainties from concentration measurements, as explained in Section 5.3 and Section S4.1 in Supplementary Information. We acknowledge that additional data would be valuable in further reducing and characterizing uncertainties. Additionally, we have added sensitivity analyses to show the dependencies of the frag table coefficients on the organic nitrate concentrations for our ambient data in response to RC1-4 below.

**RC1-2**. Section 4.2: It is not clear why geometric mean was used to derive the ratio of pure organic nitrate ($R_{pON}$). Not being a statistician, I do not understand the full reasoning behind using geometric mean, and why it makes more sense than arithmetic mean. If the authors could provide more details and references why geometric mean between two extreme values was used would strengthen the selection and section.

We currently only have two references for $NO_x^+$ ratio of pure organic nitrate ($R_{pON}$) for capture vaporizer (CV) instrument data; one being from this study (upper limit and lower limit from glyoxal chamber experiment) and the other being from Hu et al., 2017. Since we see the tendency of m/z 46 (and thus NO3_46 or $NO_2^+$) being produced in the vaporizer in relatively small quantity compared to NO3_30 (or $NO^+$), the tendency for $R_{pON}$ is therefore to approach zero value. We assume that if there were similar future studies determining $R_{pON}$ in CV instruments, the central value would also be more likely to fallcloser to zero. The geometric mean would be able to mimic this central tendency using the two extreme $R_{pON}$ values better than using the arithmetic mean. Moreover, it is common to use geometric means when determining averages for ratios, which emphasizes the importance of relative vs absolute changes in its effect on the equations it is used in here. Thus, we consider it to be the more conservative approach, since we do not have sufficient data to assess the type or degree of normality of the statistical distribution.

We have revised a sentence and completed the information to the manuscript:

**L291-294:**

> **Since we see the tendency of m/z 46 signal intensity (and thus $NO_2^+$) to be produced in the vaporizer in relatively small quantities compared to $NO^+$, the tendency of $R_{pON}$ therefore is also to approach zero (non-normal distribution). With only limited information about $R_{pON}$ in CV unlike SV, we use the geometric mean instead of arithmetic mean to establish the expected central value of $R_{pON}$. We note that it is common to use geometric means to estimate averages of ratios.**

**RC1-3**. Section 6.1: Co-located measurements of pON is extremely challenging and rarely possible, which is understood. If there was anyway to have a co-located measurement, from a chamber study or somewhere else where there was another ACSM with CV and another pON, would strengthen this section. Currently, the results shown in Figure 5 and 6 are hard to judge if the trends and mass concentrations make sense.

The authors agree to the statement of the reviewer. It is the reason why in the "Conclusion and recommendations" section, we encourage other CV-ACSM and CV-AMS users to conduct similar studies with their instruments (e.g., measuring various pure organic nitrate, co-located measurements of CV-ACSM and CV-HR-AMS) to obtain more precise numbers to analyze organic nitrate concentrations.

**RC1-4**. Section 6.2: This section is not very convincing in that the TOF-ACSM CV is sensitive towards pON. Combination that the fraction of pON reported by AMS and ACSM diverge, indicating that a single correction value for the fragmentation table may not be applicable, and that the scatter plot (Figure 8) is really driven by two points (e.g., the values before limonene was injected, which is ~0, and the values after limonene was injected, which could be averaged into one point). Thus, the analysis from this one chamber experiment is suggestive that the CV TOF-ACSM may not be able to quantify pON and would potentially over attribute nitrate signal to pON instead of ammonium nitrate (by ~50-60%).

Further analysis of this one experiment, or analysis of another chamber experiment with different chemistry, if possible, is needed to better understand the uncertainty and whether it is a precursor dependency and/or uncertainty with a constant fragmentation table correction.

To address this comment and clarify the uncertainties, we have added new analysis to the paper. We explore how the correction for the fragmentation table affects the fraction of pON by performing additional sensitivity analysis to the nitrate pollution episodes. We choose to show this rather than another chamber experiment because we have less dependence on a specific precursor in the complex ambient air. The sensitivity analysis is presented in Figure R3 below, where we show how the calculated pON concentration is affected by changing $R_{pON}$, $a_{Org[30],[29]}$, and $a_{Org[46],[45]}$.

We found that changing the $a_{Org[46],[45]}$ value (we use values listed in Table S4 and Figure S1d) leads to substantial changes in computed pON concentration, which is not the case for $a_{Org[30],[29]}$ (we use values listed in Table S4 and Figure S1c). A similar result was found when we varied $R_{pON}$, where values from $10^{-3}$ and below do not significantly change the reported pON concentration.

This demonstrates that the limitation of this adapted $NO_x^+$ ratio method is its sensitivity towards the m/z 46 correction to obtain $NO_2^+$ signal contribution, especially when the aerosol mixture is unknown. Chemically, this suggests that the Org fragments in m/z 46 can vary substantially and comprise a substantial portion of the total m/z 46 signal. Therefore, an average correction for m/z 46 may result in a high uncertainty of calculated $f_{pON}$ using the $NO_x^+$ ratio method. This was considered in the propagation of uncertainty, where we take into account the changing $a_{Org[46],[45]}$ in the reported $f_{pON}$ and therefore representing the reality better.

**We will include Figure R3 as Figure 7 (L414) in the updated manuscript, and the previous** Figure 7 **becomes Figure 8 (L420) due to renumbering.**

[Figure]

**Figure R3 (Figure 7). Sensitivity analysis of (a) $R_{pON}$, (b) $a_{Org[30],[29]}$, and (c) $a_{Org[46],[45]}$ to the pON concentration ($NO_3$ in pON) calculated using adapted $NO_x^+$ ratio method. The time series in each case is an ambient pollution episode in Cabauw, the Netherlands, during spring, summer, autumn, and winter period (i-iv). The results show that (a) $R_{pON} \leq 10^{-3}$ does not show significant differences in reported pON concentration, and (b) the reported pON concentration is not sensitive to the change of $a_{Org[30],[29]}$. In contrast, the results (c) show a significant change in reported pON concentration when $a_{Org[46],[45]}$ is varied, showing that this correction is the primary limitation of the $NO_x^+$ ratio method in CV-ACSM, because it can be highly dependent on the calculation of $NO_2^+$ signal contributions to m/z 46.**

We added new paragraphs to the Section 6.1 as follows:

- Section 6.1 title changed from 'Trend of $f_{pON}$ and $f_{pAmN}$ vs. ACSM Org and NH$_4$ at rural site' to 'Ambient measurements at rural site' since we also include sensitivity analysis to the ambient measurements.

- L401-416:

    Unless there are co-located ambient measurements of UMR and HR instruments, the reported concentration of NO$_3$ in pON depends on the value of $R_{pON}$ and multipliers used to calculate NO$^+$ and NO$_2^+$ signal contribution in the fragmentation table. The sensitivity of these variables needs to be assessed to understand which parameter is the most critical in the separation of inorganic and organic nitrate signal from ACSM.

    In Fig. 7, we show the sensitivity analysis of $R_{pON}$, $a_{Org[30],[29]}$ and $a_{Org[46],[45]}$. We varied $R_{pON}$ from zero to $R_{pON}$ = 0.0072 (calculated using RoR = 3.29). Fig. 7a suggests that for $R_{pON} \leq 10^{-3}$, the reported pON concentrations are not significantly different and therefore confirm the lower limit of $R_{pON}$ approaching zero, as established in Table 2. The value of $R_{pON}$ calculated using RoR (in this case $R_{pON}$ = 0.0072) shows relatively higher pON concentration, which is consistent with its use as the upper limit of $R_{pON}$.

    We also varied $a_{Org[30],[29]}$ and $a_{Org[46],[45]}$ using the values listed in Table S4 and Fig. S1c,d in SI. Fig. 7b shows that the calculated pON concentration is sensitive to $a_{Org[46],[45]}$, which is not the case for $a_{Org[30],[29]}$. It further demonstrates that the limitation of this adapted NO$_x^+$ ratio method is its sensitivity towards $a_{Org[46],[45]}$ to obtain NO$_2^+$ signal contribution. Chemically, this suggests that the organic contribution in m/z 46 can vary and comprises a substantial portion of the total m/z 46 signal. Therefore, an average correction for m/z 46 may result in a high uncertainty of calculated $f_{pON}$ using the NO$_x^+$ ratio method. This was considered in the propagation of uncertainty, where we take into account the changing $a_{Org[46],[45]}$ in the reported $f_{pON}$, therefore representing a range of the observed organic nitrate contribution.

Furthermore, we decided to add a second validation of the adapted NO$_x^+$ ratio method using PMF analysis to the CV data. With this, we can show how the NO$_x^+$ ratio the method in CV measurements responds to the change of nitrate composition when we inject an organic precursor, compared to the NO$_x^+$ ratio method in SV measurements and the PMF analysis of the CV data. **Since we want to spotlight the instrument response to organic nitrate formation upon limonene injection between CV-ACSM and SV-AMS rather than the magnitude (CV/SV), we made some updates related to this:**

- **Removal of** Figure 8 (in preprint), **showing the scatter plot of $f_{pON}$ from CV vs. SV. Instead, we will only report the r$^2$ values which will be shown later in this response.**
- **We remove the CV/SV ratio in the abstract and rather include the sensitivity analysis to show uncertainties from CV** (more detailed below)
- **Abstract**, L11-13 → **L13-16:**
    A comparison to a high-resolution time-of-flight aerosol mass spectrometer equipped with a standard vaporizer (SV-HR-ToF-AMS)  **and positive matrix factorization (PMF) method shows similar response of increasing** particulate organic nitrate fraction  **with uncertainties**

**mainly from sensitivity to fragmentation table correction when obtaining $NO_2^+$ signal.**

- **Section 6.2,** L396-398 → **L450:**

  …(62 g mol$^{-1}$/46 g mol$^{-1}$) to recalculate the pON mass. ~~For the SV-AMS, pON represents ~32% of the nitrate mixture when reported as −ONO₂. Without that correction, the measured pON would be underestimated. For the CV, f~pON~ represents on average ~50% of the total pNO₃ concentration, higher than the SV (ODR fit slope CV/SV = 1.59, r² = 0.92; see Fig. 8).~~ There...

Before going to the PMF analysis, we would like to give some updates regarding some changes we made to the SV-AMS data.

1. We reanalyzed the SV-HR-ToF-AMS data and found that the ion $CH_2O_2^+$ was not fit to the m/z 46. This has resulted in the overestimation of $NO_2^+$ ion signal, for both the chamber experiment and the calibration of $NH_4NO_3$ that we use to determine $R_{pAmN}$. With this, $R_{pAmN}$ is found to be 0.61 instead of 0.68.
   However, since the fit of $CH_2O_2^+$ remains quite constant throughout the experiment and we use RoR to calculate $R_{pON}$, this does not change the $f_{pON}$ value calculated in the chamber as every value moves to the same direction.
2. We omit the use of RoR = 2.75 ± 0.41 (~15% uncertainty) to determine the upper and lower limit of $R_{pON}$ for SV-HR. This value is suggested in Day et al. (2022) and more suitable for complex mixtures of organic nitrate in the atmosphere. The uncertainty of the RoR for chamber experiment is likely higher (~25% uncertainty, from standard deviation of the dataset used to obtain the RoR in Day et al. (2022)) because about the data are from a single precursor. Therefore, we now use **RoR = 2.75 ± 0.70** to calculate $R_{pON}$ for the SV-HR analysis. As a result, we have $R_{pON}$ = 0.18; 0.22; 0.30 as the upper limit, mean, and lower limit of $R_{pON}$, which means the uncertainty of $f_{pON}$ is now larger.

Here, we show the updates related to the SV-AMS data in the manuscript.

- L382-384 → **L425-427:**

  **… For the AMS instrument, the measurements of pure pAmN give $R_{pAmN}$ = 0.61 ± 0.05 and the $R_{pON}$ value is calculated using RoR = 2.75 ± 0.70 (Day et al., 2022a), which gives $R_{pON}$ = 0.18; 0.22; 0.30 as lower limit, mean, and upper limit, respectively.**

- **The values are also updated accordingly as shown in Figure R4 below, which appear in the updated manuscript as Figure 8, L416 (before was** Figure 7 in the preprint). Figure R4 contains the time series of $f_{pON}$ obtained from PMF method, which will be described below.

[Figure]

**Figure R4 (Figure 8). (a,b) The time series in 2 min time averaging of R_obs, R_pAmN, and R_pON measured by CV-UMR-ToF-ACSM (top) and SV-HR-ToF-AMS (middle). The fragmentation table specific for terpene is used to obtain the C_NO+ and C_NO2+ of the chamber experiment. (c) The time series in 2 min time averaging of f_pON from NO_x^+ ratio method applied to SV-HR-ToF-AMS and CV-UMR-ToF-ACSM, as well as PMF method applied to CV-UMR-ToF-ACSM. The markers represent geometric mean for NO_x^+ ratio method applied to CV-UMR-ToF-ACSM (circle), mean for NO_x^+ ratio method of SV-HR-ToF-AMS (square) and for PMF method of CV-UMR-ToF-ACSM (cross). The whiskers represent the uncertainties from the value range of R_pON combined with the uncertainties from electronic noise, ion counting statistics, fragmentation table (for UMR), and R_pAmN. The uncertainty from the PMF analysis is not shown for simplicity.**

We also performed a PMF analysis to the CV-ACSM dataset of period highlighted in red square **(Figure R5, which will be put in the Supplementary Information as Figure S2). The details of the PMF analysis are as follows, which will be included in the main article (Section 6.2) and Supplementary Information (Section S5).**

- We describe briefly the PMF analysis we performed to the CV-ACSM data **(Section 6.2, L428-436):**

  **We also performed PMF analysis using the ACSM data including OA, NO_x^+, and NH_x^+ ions, which has been similarly done in other studies (e.g. Day et al. (2022a) and references therein). The 2 min average matrices of UMR organic fragment mass spectra with a m/z of 12 to 120, fragments of ammonium (NH4_16 and NH4_17, two main signals of NH_4 which are NH_2^+ and NH_3^+), and fragments of nitrate (NO3_30 and NO3_46, two main signals of NO_3 which are NO^+, NO_2^+) are used as variables in the PMF input matrix. Fragment contributions are calculated using the terpene-related fragmentation table (see Section 3.2). We choose a two-factor solution (see Fig. S2) because we are interested in splitting the aerosol mass only into inorganic aerosol (pAmN) and organic aerosol (OA mixture, containing pON). From the organic aerosol factor, we calculate the fpON from the factor concentration time series. The details of the PMF method are described in Section S5, including the statistical summary and the diagnostic plots of the PMF analysis.**

[Figure]

**Figure R5 (Figure S2). Overview of PMF analysis to the chamber experiment. Plot (a) shows the time series of $R_{obs}$, $R_{pAmN}$, and $R_{pON}$ measured by CV-UMR-ToF-ACSM. Plot (b) shows the time series of ACSM species of $NO_3$, $NH_4$, Org, as well as the apportioned pON and pAmN stacked to fit the total $NO_3$. The red square indicates the period where the PMF analysis is performed. Plots (c-h) show the two-factor solution of PMF analysis using Org fragments from m/z 12 to m/z 120 and inorganic fragments ($NH_2^+$, $NH_3^+$, $NO^+$, $NO_2^+$) as input matrix. Plots (c-f) describe the factor profiles. F1 is shown to represent organic aerosol in the chamber, consisting of Org, nitrate and amines ($NO_2^+/NO^+ = 0.0003$), while F2 represents ammonium nitrate with negligible organic component ($NO_2^+/NO^+ = 0.0119$). Plot (g) shows the factor time series where F2 is compared with the concentration of pAmN ($C_{pAmN}$) as sum of $NO_3$ in pAmN obtained from $NO_x^+$ ratio method ($C_{NO3,pAmN}$), and an equimolar amount of $NH_4$ ($C_{NH4,pAmN}$). Plot (h) shows the factor time series where F1 is compared with the concentration of total OA as sum concentrations of Org ($C_{Org}$), $NO_3$ in pON obtained from $NO_x^+$ ratio method ($C_{NO3,pON}$), and the excess $NH_4$ that are not assigned as pAmN ($C_{NH4,excess}$).**

- **The details of the PMF analysis (Section S5, Supplementary Information):**

  **The 2 min average matrices of UMR organic fragment mass spectra with a m/z of 12 to 120, fragments of ammonium ($NH_4\_16$ and $NH_4\_17$, two main signals of $NH_4$ which are $NH_2^+$ and $NH_3^+$), and fragments of nitrate ($NO_3\_30$ and $NO_3\_46$, two main signals of $NO_3$ which are $NO^+$, $NO_2^+$) are used as variables in the PMF input matrix. Fragments of sulfate and chloride are not included because they are not added to the chamber and the concentration is found to be negligible. The values and errors of the input matrix and minimum error (minErr) were generated by Tofware v3.3 in Igor Pro 8. All fragments are calculated using the fragmentation table for terpene-related mixture (see Fig. S1).**

  **We start the PMF analysis by varying the seed value (min = 0, max = 20, delta = 1) to pick different initial values for the PMF algorithm and choose the optimum number of factors (p). After choosing p and the seed value, the rotationality of the solution is explored by varying the rotation ($f_{peak}$) value (min = -1, max = +1, delta = 0.2). Lastly,**

**bootstrapping runs are performed with 100 iterations to estimate the uncertainties in the factor profile and time series.**

**We choose a two-factor solution (see Fig. S2) because we are interested in splitting the aerosol mass only into pAmN and pON, and low residuals and local minima ($Q/Q_{exp}$) have already been reached in this configuration ($p = 2$; seed = 0; $f_{peak} = 0$). The statistical summary of the PMF analysis is presented in Fig. S3. The time series of the measured total mass, the total reconstructed PMF mass, and the total residuals, as well as the scaled residuals of each factor m/z variable of the chosen PMF analysis are shown in Fig. S4. The chosen PMF solution split the total mass concentration into F1 and F2, representing the OA mixture and pAmN, respectively (see Fig. S2c-h). F1 has a factor profile with signals mainly from organic fragments, as well as ammonium and nitrate (Fig. S2c,e), which can be assumed to be particulate organic nitrate and amines. Meanwhile, F2 profile contains mainly signals from ammonium nitrate, with negligible background organic signals (Fig. S2d,f).**

**Since we add $NO^+$ and $NO_2^+$ to the input matrix, it is interesting to see that PMF is separating the two factors based on the $NO_2^+/NO^+$. We can determine $R_{pAmN}$ from F2 (pAmN) by simply calculating $NO_2^+/NO^+ = 0.0119$ in the factor profile, which is close to the experimental value of $R_{pAmN} = 0.0115$. The same applies for $R_{pON}$, where we can use F1 (OA) that contains pON to determine $R_{pON}$, which is found to be $NO_2^+/NO^+ = 0.0003$, showing that the $R_{pON}$ value is approaching zero as expected.**

**To validate the factor profiles, the time series of F1 and F2 (see Fig. S2g,h) are compared to the ACSM Org, $NO_3$, and $NH_4$ time series. The concentration of F2 is compared to the total concentration of pAmN ($C_{pAmN}$), which is the total concentration of $NO_3$ in pAmN ($C_{NO3,pAmN}$, obtained from $NO_x^+$ ratio method) and an equimolar amount of $NH_4$ ($C_{NH4,pAmN}$). The comparison suggests a good correlation between the two ($r^2 = 0.98$). Similarly, the concentration of F1 is compared to the total concentration of OA ($C_{OA}$), which is the total concentration of organic aerosol in the chamber, assumed as the sum of concentrations of total ACSM Org ($C_{Org}$), $NO_3$ in pON ($C_{NO3,pON}$, obtained from $NO_x^+$ ratio method), and the excess $NH_4$ that has not been assigned to ammonium nitrate ($C_{NH4,excess} = C_{NH4,total} - C_{NH4,pAmN}$), which is relatively small compared to the total ammonium. The comparison also shows a good correlation between the two ($r^2 = 0.99$).**

**In order to be able to compare $f_{pON}$ from the PMF analysis with other methods, we calculated $f_{pON}$ from F1. Since $NO_x^+$ fragments account for ~48% of F1 profile, $C_{NO3,pON}$ will have such contribution to the concentration of F1. By taking the ratio of $C_{NO3,pON}$ to the total concentration of $NO_3$, we can calculate $f_{pON}$ from PMF. The comparison of $f_{pON}$ obtained from PMF analysis and from $NO_x^+$ ratio method (both CV-UMR-ToF-ACSM and SV-HR-ToF-AMS) can be seen in Fig. 8.**

- **Then, we integrated the PMF method results into the existing manuscript as follows, in Section 6.2,** L401-408 → **L453-475:**

Both **$NO_x^+$ ratio methods (**SV-AMS and CV-ACSM**) and the PMF method (CV-ACSM)** show a similar response to the injection of limonene, whereupon $f_{pON}$ increases rapidly from **~0 to ~0.3-0.5**. The agreement between the two instruments on this initial pON production is encouraging. However, after this initial jump,  **the trend of fpON seems to**

vary. While mean $f_{pON}$ calculated using $NO_x^+$ ratio method from CV-ACSM remains steady after the injection, the other two results show a gradual decrease of $f_{pON}$ as the chamber dilutes.

The $NO_x^+$ ratio method on CV-ACSM data shows a similar change in $f_{pON}$ relative to the PMF method immediately after limonene injection but then continues to decrease over time (see Fig. 8). The PMF method combines the variations of Org, $NO_x^+$, and $NH_x^+$ ions to obtain the factor profiles, therefore allowing a more subtle change in the chamber composition to be taken account. The $NO_x^+$ ratio method in CV-UMR, in contrast, only takes into account fragments that are in m/z 30 and m/z 46 using a constant fragmentation table relationship.

Similarly, the SV-AMS also shows a gradual decrease of mean $f_{pON}$, unlike the $NO_x^+$ ratio method from CV-ACSM (see Fig. 8). This suggests that changing contributions of organics at m/z 30 and m/z 46  may be taken into account by HR peak fitting  and not by the UMR fragmentation table causing a divergence as the chamber aerosol dilutes. Based on the sensitivity analysis in Fig. 7, the signal contribution of $NO_2^+$ (m/z 46) is the largest source of uncertainty in the CV-ACSM, since the adapted method is sensitive to the change of $a_{Org[46],[45]}$. On the other hand, the uncertainty of RoR used to calculate $R_{pON}$ accounts for the largest contribution to the uncertainty calculated for SV-AMS. Since the RoR in Day et al. (2022a) relies on the average value of a broad range of organic nitrate, a chamber experiment that uses a specific precursor is likely to have $R_{pON}$ further away from the average value than, for instance, a complex ambient mixture.

 Nevertheless, we are encouraged by the match in responses upon formation of organic nitrate, indicating that the $NO_x^+$ ratio method is similarly sensitive to changing nitrate speciation in both instruments. **When considering the propagation of uncertainty, we observe overlaps between the results of CV-ACSM and SV-AMS. Further investigation of the detailed response of each instrument to changing aerosol composition would be valuable.**

- **The statistical summary and diagnostic plot of the PMF analysis are shown in Figure R6 and Figure R7, which are included in Supplementary Information as Figure S3 and Figure S4.**

[Figure]

**Figure R6 (Figure S3). Diagnostic plots of PMF analysis showing (a) Q/Q$_{exp}$ vs. number of factors (p), (b) Q/Q$_{exp}$ vs. seed value, (c) Q/Q$_{exp}$ vs. f$_{peak}$ value, and (d) correlation of time series and mass spectra among two PMF factors (R time series vs. R profiles). The value of p = 2, seed = 0, and f$_{peak}$ = 0 are chosen.**

[Figure]

**Figure R7 (Figure S4). Diagnostic plots of the chosen PMF solution showing (a) time series of the measured total mass and reconstructed PMF mass, (b) time series of residual and scaled residual of the least-square-fit, (c) distribution of scaled residuals for each organic fragment m/z, and (d) distribution of scaled residuals for each inorganic fragment.**

Finally, the **conclusion,** L442-445 → **L513-517:**

… and organic aerosol composition, respectively. ~~Co-located high-resolution time-of-flight aerosol mass spectrometry equipped with standard vaporizer (SV-HR-ToF-AMS) and RoR to determine RpON in a chamber experiment shows a good correlation (r2 = 0.92) of fpON with CV-UMR-ToF-ACSM observation, with the latter estimating ~1.6 times higher fraction than the former.~~ **The adapted NO$_x^+$ ratio method applied to CV-UMR-ToF-ACSM measurements in a chamber experiment is able to replicate the response to precursor injection observed from PMF analysis of the same measurements, as well as a co-located high-resolution time-of-flight aerosol mass spectrometer equipped with standard vaporizer (SV-HR-ToF-AMS). The largest uncertainties in this comparison come from a$_{Org[46],[45]}$ (CV-UMR-ToF-ACSM) and RoR (SV-HR-ToF-AMS).** The adapted NO$_x^+$ …
* * *
**Response to RC2 (Anonymous Referee #3):** https://doi.org/10.5194/amt-2024-191-RC2

The manuscript presents a detailed analysis of the $NO_x^+$ ratio method for apportioning organic and inorganic nitrate aerosols using a capture vaporizer (CV) in the Aerosol Chemical Speciation Monitor (ACSM). The study involves three main components: revising the fragmentation table to enhance nitrate detection, evaluating the NO2/NO ratio for source apportionment, and applying this method to both real-world and chamber data. With more and more CV-ACSM being used, the clarification on the organic/inorganic nitrate will improve the understanding of this topic. In general, the manuscript is well-written and organized. After reading through the whole manuscript, I outline several concerns, along with suggestions for improvement.

**RC2-1**. The revised table's performance should be tested in various settings to ensure its broad applicability. The authors obtained the slope for the fragmentation table to correct nitrate at m/z 30 and 46. However, the author never showed the verification of this method with real ambient data. E.g. Hu et al. (2017) demonstrated a better agreement on the total nitrate between SV and CV after UMR fragmentation revision in a biogenic-dominated area. How about this revision in the urban or rural areas, as well as the chamber studies?

The verification of the revised table's performance for CV instruments in ambient measurements is possible when there are co-located measurements using a CV-UMR instrument and an HR instrument. The HR instrument can be equipped with either SV or CV, but generally CV is better to have direct comparison (the table corresponds to ion formation happening in the vaporizer).

Unfortunately, it is very rare to have both instruments at the same time and place, and we have not yet been able to perform this verification. Therefore, we are unable to provide any real urban vs rural ambient measurements. Instead, we rely our verification based on re-applying the revised table to the total m/z 30 and m/z 46 from CV-HR spectra that has been "degraded" into CV-UMR spectra (mix nitrate + organic contribution) to simulate as if we measure using CV-UMR instrument. Then, we check the agreement between the "predicted" organic contribution (obtained from fragmentation table calculation) vs the direct sum of organic contribution in each m/z measured by the HR instrument. The CV-HR spectra are available in the Capture Vaporizer AMS Spectral Database (Hu et al. URL: https://cires1.colorado.edu/jimenez-group/AMSsd_CV/; Hu et al. 2018a; Hu et al. 2018b).

This effort is shown in Figure 1c,d in the manuscript. It may not be ideal since we are re-applying the revised table to the data where we get the revision itself. However, this database includes a variety of datasets from different chamber studies, real ambient measurements (as positive matrix factorization factor), and laboratory standard, so we assume that the performance represents a broad range of conditions.

In contrast to Hu et al. (2017) who compares the total nitrate to check the agreement between two measurements, we use the Org_30 (predicted UMR vs measured HR) as well as Org_46. This is done since the correlation of Org_30 to Org_29 and Org_46 to Org_45 are independent from $NO_3$, and there are HR spectra included in the dataset that contains no nitrate fragments.

The improvement of the new revision to the fragmentation table is shown in Table S4, where we perform what is done to Figure 1c,d to various fragmentation table corrections from different studies (the default from Allan et al. (2004), Fry et al. (2018), and Hu et al. (2017)). The use of revised fragmentation table shows a good agreement (via the value of predicted-to-measured ratio and $r^2$ value) for calculated Org_30 (multiplier $a_{Org[30],[29]} = 0.311$) compared to the default fragmentation table from Allan et al.

(2004) and Fry et al. 2018. This multiplier is actually similar to the numbers obtained for biogenic mixture reported by Hu et al. (2017) (0.31-0.32), which is also shown performing well for the same dataset.

**RC2-2**. Are there more RoR ratios reported in the literature literature? In SV, Doug et al. (2022) checked a variety of literature to determine the final RoR ratio.

As far as we know, there have been only two studies trying to determine the $NO_x^+$ ratio of pure organic nitrate ($R_{pON}$) from CV instrument (this study and Hu et al. (2017)). RoR was not calculated in Hu et al. 2017, but we use their measurements to calculate an RoR. By submitting this work, we hope to encourage other people to report measurements from their CV instrument as well.

**RC2-3**. Potential nitric acid formation under high humidity should be addressed to ensure accurate results. Line 250, the authors conducted the chamber study at RH around 90% with $NO_3$ radical. There will be a formation of nitric acid. Note that even in the absence of $NH_3$, nitrate acid can still be formed under high RH with $N_2O5$. How did the authors exclude the inferences of nitric acid to the $NO_2^+/NO^+$ ratio determined by organic nitrate in this study?

Our experiment was run in excess $NH_3$ concentration which immediately converts $HNO_3$ into $NH_4NO_3$. Therefore, the interference of $HNO_3$ should not be a concern in this analysis. **We address this information into the manuscript under Section 6.2.**

**L441-443:**

> **Although nitric acid ($HNO_3$) can be formed by $N_2O_5$ hydrolysis under these humid conditions, experiments were run in excess $NH_3$ and therefore we expect no substantial increase of $HNO_3$ that can affect $NO_x^+$ ratio.**

**RC2-4.** The applicability of the $NO_x^+$ ratio method at specific concentration levels should include considerations of organic nitrate fractions and averaging times. In Section 5.2, the authors declared the $NO_x^+$ ratio method can be used for total nitrate concentration at 0.6 µg m$^{-3}$ and above, which is misleading. The sole detection limit of $NO^+$ cannot determine the limited usage of this method. I think Figure 4 gives a more comprehensive overview of this method. The usage of the $NO_x^+$ ratio method was also limited by the fraction of pON in total nitrate and averaging time; this should be mentioned clearly in the main text, as well as shall be revised in the abstract and conclusion. In the urban area, where the $f_{ON}$ accounts for 10% or even less, the CV-AMS at 10 min resolution cannot be used.

It is correct that the limit of total nitrate concentration at 0.6 µg m$^{-3}$ only applies when the time averaging is 120 minutes since it depends on the time averaging, as the referee mentioned. These values reflect the concentration cut-off due to the data filtering rather than the concentration limit for reliable results.

We tried the referee's suggestions to look for the limitations of $NO_x^+$ ratio method for a given fraction of pON in total nitrate by replicating the Figure 4 for the absolute uncertainty of fpON ($s_{fpON}$) against pON fraction in total nitrate ($f_{pON}$), as shown in Figure R8b below. We can consider the data point above the one-to-one line to be not reliable since its absolute uncertainty is larger than the calculated $f_{pON}$ itself. We can observe that the limit mostly falls between 12% to 20% $f_{pON}$ as pointed out by the referee (roughly 20% for 10 min resolution, and 12% for 120 min resolution).

*RC2-6 (moved up here as the reply is related to RC2-4).* Line 350, I do not understand why authors only point out a fraction of 17%. In most of the urban areas, the pON fraction in total nitrate is less than 15%.

We thank the referee for pointing this out. On top of the 17% fraction of pON reported by Yu et al. 2024 in Shenzhen, China, we will complete the information in our manuscript by mentioning the finding of 13% fraction of pON in urban Barcelona, Spain related to the work of Mohr et al. 2012 and Pandolfi et al. 2014, and also Xu et al. 2021 who reported 9.8% fraction in wintertime Beijing, China. Since now 9.8% fraction reported by Xu et al. (2021) is the lowest pON fraction we have, we replotted Figure 4 as **Figure R8**a below where we set the absolute uncertainties <10% (<0.1) to show the concentration limit. It means that now the concentration limit of $pNO_3$ that gives us <10% absolute uncertainty of $f_{pON}$ varies from ~2 μg m$^{-3}$ at 120 min time averaging, to ~10 μg m$^{-3}$ at 10 min time averaging. Therefore, in order to obtain reliable results with minimum uncertainty using this method, the nitrate concentration have to be within 2-10 μg m$^{-3}$ limit and the organic nitrate fraction within 12-20%.

**We will use Figure R8 to replace Figure 4 in the Section 5.3, and mention the $f_{pON}$ limit together with the concentration limit to report uncertainties <0.1 or <10% for the method, instead of uncertainties <0.2 (we now use 9.8% fraction reported by Xu et al. (2021) as the lower range).**

- **Abstract,** L9-10 → **L9-12:**

  **"Data pre-treatment filters concentrations of particulate nitrate below 0.6-2.0 μg m$^{-3}$, depending on the time averaging. The method detection limit, when considering ±10% absolute uncertainty of organic nitrate fraction, is found to be 2 μg m$^{-3}$ (120 min averaging) to 10 μg m$^{-3}$ (10 min averaging) for total particulate nitrate concentration and 10% (120 min) to 20% (10 min) for organic nitrate fraction."**

- **Section 5.2,** L326-332 → **L346-350.**

  **"The combination of data filtering and time averaging shows different concentration cut-off for calculation of the $NO_x^+$ ratio. The concentration cut-off is lower for longer averaging times due to the improvement of $C_{DL,NO2+}$. For measurements in this study, the $C_{pNO3}$ cut-off for 10 min, 30 min, 60 min, and 120 min time averaging are 2.0, 1.2, 0.9, and 0.6 μg m$^{-3}$, respectively. Because there is a trade-off between time resolution and the concentration cut-off, for a given dataset, the timescale of typical variations should be assessed in order to determine the appropriate averaging time."**

- **Section 5.3.:**

  o **We also renamed Section 5.3 from "Error propagation" to "Propagation of uncertainty" to more accurately reflect the contents of this section.**

  o **Figure R8 replaces Figure 4, including the caption, in the updated manuscript.**

[Figure]

**Figure R8 (Figure 4). The chemical coordinate plot (quantile average) between (a) $s_{fpON}$ and $C_{pNO_3}$ (logarithmic scale), and (b) $s_{fpON}$ and $f_{pON}$ (linear scale), with $R_{pAmN} = 0.0237$ as filter $NO_x^+$ ratio at various averaging of the time series. The line and marker trace represents the average uncertainty produced from the geometric mean of $f_{pON}$. The uncertainty consists of uncertainties of ion counting statistics from measurements, uncertainty from ODR fit slope of fragmentation table correction, and uncertainty of $R_{pAmN}$. The colored shading represents the standard deviation of each quantile, while the whisker is the standard error. The shading and whisker both include the uncertainty of $R_{pON}$ coming from the lower and upper limit of $R_{pON}$ ($R_{pON} = 0.0001$ and $R_{pON} = 0.0072$), and also the uncertainty of the average quantile. Uncertainties of $f_{pON}$ <0.1 (absolute value) is reached at $pNO_3$ concentration >10 μg m$^{-3}$ for 10 min time averaging, while at 60 min, it is reached already at ~4 μg m$^{-3}$. In terms of fraction, uncertainties of $f_{pON}$ below the calculated $f_{pON}$ ($s_{fpON} < f_{pON}$) is reached at $f_{pON} \sim 0.2$ for 10 min averaging, while at 60 min, it is reached at $f_{pON} \sim 0.17$.**

○ L350-353 → **L368-380:**

> **Several studies reported $f_{pON}$ lower than 20%, which occur mainly in urban areas and during a colder period. Yu et al. (2024) observed a lower range of annual average of urban $f_{pON}$ in China to be ~17%, while Mohr et al. (2012) and Pandolfi et al. (2014) reported ~13% fraction in Barcelona, Spain, and Xu et al. (2021) reported 9.8% fraction in wintertime Beijing, China. If we use the lower range of $f_{pON}$ of ~10% as reference for the minimum uncertainty needed to report reliable $f_{pON}$, we can observe that the lowest $pNO_3$ where we obtain below 0.1 absolute uncertainty in $f_{pON}$ decreases along with time averaging as well. Uncertainties below 0.1 can only be reached at $pNO_3$ concentration higher than 10, 7, 4, 2 μg m$^{-3}$ at 10, 30, 60, and 120 min time averaging, respectively.**

> **Figure 4b shows the relationship between the absolute $s_{fpON}$ and $f_{pON}$. The limit at which the absolute value of $s_{fpON}$ is below or equal to $f_{pON}$ (minimum uncertainty) is found to be 20%, 15%, 14%, 12% at 10, 30, 60, and 120 min time averaging, respectively. This result suggests that the $NO_x^+$ ratio method in CV-UMR-ToF-ACSM is more reliable to analyze nitrate pollution episodes or chamber experiments, and not for low background $pNO_3$ concentrations. By combining both the concentration limit and the fraction limit, we suggest that in the region where $pNO_3$ concentration is <10 μg m$^{-3}$ and/or $f_{pON}$ <12%,**

the method requires a longer time average to calculate $f_{pON}$ to achieve minimum uncertainty.

- **Conclusion,** L432-438 → **L499-509:**

  …is  **able to filter unreliable measurements** with  **concentration cut-off ranging from 0.6 to 2.0 µg m$^{-3}$**, depending on time averaging. **This data pre-treatment filters data points with high fraction uncertainty (above ±0.5) and decreases the average uncertainty by √N for each N-fold of averaging from 10 min.**

  **With a longer time averaging, the concentration limit and fraction limit improve, which allows more reliable determination of $f_{pON}$ and $f_{pAmN}$. The method reports absolute uncertainty of particulate organic nitrate <10% at the total particulate nitrate concentrations of 2 µg m$^{-3}$ (120 min time averaging) to 10 µg m$^{-3}$ (10 min time averaging) and organic nitrate fraction of 10% (120 min time averaging) to 20% (10 min time averaging).** We recommend users to average the time series to 30 min or 60 min to retain information about real ambient variation, while improving the reliable nitrate concentration limit. This may also be convenient when comparing to auxiliary data that are typically reported half-hourly or hourly. **In the region where pNO₃ concentration is <10 µg m$^{-3}$ and/or $f_{pON}$ <12%, longer time averaging may be necessary to achieve the absolute uncertainty <10%.** In studies where noise…

**RC2-5.** Have the authors tried to determine the detection limits of $NO_3$ using the $NO_x^+$ ratio using the method below, which is obtained in Figure 2 in Hu et al. (2017)? Their work is for HR nitrate. Will this method lead to similar detection limits of total UMR nitrate with what was obtained in this study?

[Figure]

We did not explore the determination of $NO_3$ detection limit using the $NO_x^+$ ratio since, unlike HR, we do not obtain $NO^+$ and $NO_2^+$ signal directly from UMR measurements. Therefore, we determine it using the procedure described by Aerodyne, to measure the filtered ambient air for a certain amount of time and taking the three times of standard deviation of the noise as detection limit. From this method, we found that the detection limit for $NO_3$ is ~240 ng m$^{-3}$ for ACSM-UU and ACSM-RUG (value converted to 1 min time averaging to allow direct comparison), which is comparable to ~100 ng m$^{-3}$ (1 min time averaging) mentioned in Hu et al. (2017).

**RC2-6.** Line 350, I do not understand why authors only point out a fraction of 17%. In most of the urban areas, the pON fraction in total nitrate is less than 15%.

Reply to RC2-6 is merged with the reply to RC2-4 above.

**RC2-7.** There are too many acronyms. A summary of the abbreviations and their corresponding full names in an appendix table improve manuscript accessibility.

**Thank you for the suggestion. We have added an appendix table to list all terms and abbreviations used in this manuscript as Appendix A.**
* * *
**Response to RC3 (Anonymous Referee #4):** https://doi.org/10.5194/amt-2024-191-RC3

In their manuscript "Development and validation of a $NO_x^+$ ratio method for the quantitative separation of inorganic and organic nitrate aerosol using CV-UMR-ToF-ACSM", Nursanto and co-authors present and evaluate a method to extract quantitative information on the fractions of organic and inorganic particulate nitrate from unit-mass resolution data from a ToF-ACSM, equipped with a capture vaporizer. For this purpose, they analyze the ratio of the fragments at m/z 30 ($NO^+$) and m/z 46 ($NO_2^+$), which is different for inorganic and organic nitrates. Since the capture vaporizer generates stronger fragmentation and consequently less m/z 46 signal, compared to that at m/z 30 and since this brings the m/z 46 signal closer to the limit of detection and requires an improved correction for "other" contributions to the nitrate-related m/z, the method to extract inorganic and organic nitrates from AMS mass spectra needs improvements and extensions. The method, developed by the authors, is clearly described in their manuscript and several validation experiments are presented.

The manuscript is clearly written and the developed method is clearly described with sufficient detail. The validation experiments and analyses are also clearly described and good to follow. The presented method is valuable for the growing ACSM aerosol monitoring community, providing a method to separate organic and inorganic nitrates in their data sets. The description and validation of this method fits well into the scope of Atmospheric Measurement Techniques.

The manuscript shows a number of (minor) technical issues, which should be addressed before publication. In addition, my major concern is, whether the limitations and uncertainties of the method are fairly addressed. With uncertainty ranges of frequently 100% and above, the method does not necessarily provide robust and always meaningful information on the quantitative contributions of organic and inorganic nitrate to total nitrate. I do not think that this limitation is adequately addressed in the manuscript. Please see my detailed comments regarding this issue. I think, after addressing these issues, the manuscript should be published in AMT.

**RC3-1.** L9: Providing a number for the concentration limit (0.6 µg/m$^3$) only makes sense when the associated averaging time is also provided here (same comment for line 65/66).

Please see responses to related comments RC2-4 and RC2-6 above.

**RC3-2.** L12: I doubt that "good" is the right word to describe this correlation. While the correlation is tight (high r$^2$), the CV finds almost 60% higher pON fractions, compared to the SV.

The authors agree with the referee. Our main goal including the chamber experiment is to show that the adapted $NO_x^+$ ratio method is sensitive enough to the change in nitrate composition. However, the results only show a tight correlation and not the magnitude. Since we are using a very specific chamber experiment to validate the method applied to CV-UMR-ACSM measurement, it is also prone to bias of a specific chemistry in the chamber, which is not the main focus of the manuscript.

Please see the response to comment in RC1-4 by above. We have conducted additional analyses to respond to this concern.

It would be informative to the reader not only to provide lower nitrate concentration limits but also information on accuracy and uncertainty of this method in the abstract.

The reply to this comment is related to RC2-4 and RC2-6 above. We have added information on accuracy and uncertainty of this method in the abstract.

**RC3-3.** L13: This sounds like that the presented method is universally usable for these instruments. It would be desirable that a statement is included in the abstract, stating that for each instrument (and potentially even tuning-specific conditions of the instrument like aerosol beam alignment or vaporizer temperature) the method has to be adapted.

We modify the sentence as follows:

L13-15 → **L16-18:**

> We propose that researchers use this $NO_x^+$ ratio method for CV-UMR-ToF-ACSM **(adapting the appropriate fragmentation table and data pre-treatment for each specific application)** to quantify the particulate organic nitrate fraction at existing monitoring sites…

**RC3-4.** L38-39: I suggest rephrasing this sentence to make clearer that the combination of the AMS vaporizer and ionizer interactions with the analytes results in different fragmentation patterns, i.e., stress the process that leads to the fragmentation pattern instead of presenting it as an inherent feature of the different nitrates.

Generally, the $NO_x^+$ ratio is not just different between inorganic and organic nitrates, but between nitrates with different volatility, e.g., between ammonium nitrate and other, less volatile, inorganic nitrates (e.g., $KNO_3$), which also show a larger $NO_x^+$ ratio.

We modify the sentences as follows.

L38-40 → **L42-45:**

> The basis of the $NO_x^+$ ratio method comes from **the different fragmentation patterns of chemical species due to the interaction of the mass spectrometer's vaporizer and ionizer with the analytes.** The empirical observation shows that nitrates attached to an organic moiety have different fragmentation patterns **compared to nitrate in the form of $NH_4NO_3$, and also other less volatile inorganic nitrate.** Thus, **each nitrate** will have different $NO_x^+$ ratios, …

**RC3-5.** L42 (Eq. 1): Also introduce "C" in the text.

We introduce $C_{NO+}$ and $C_{NO2+}$ under the equation.

L42-43 → **L47-50:**

> $$R_\nu = \frac{\left(C_{NO_2^+}\right)_\nu}{\left(C_{NO^+}\right)_\nu} \tag{1}$$
>
> ν: nitrate compound or mixture measured
>
> $C_{NO_2^+}$**: signal intensity of $NO_2^+$**
>
> $C_{NO^+}$**: signal intensity of $NO^+$**

**RC3-6.** L50: ACSM means aerosol chemical speciation monitor (see Aerodyne website).

The sentence is modified into as follows.

L50 → **L57:**

> The aerosol chemical speciation **monitor** (ACSM; Aerodyne Inc.) is a unit-mass resolution (UMR)…

**RC3-7.** L58-59: While the CV is actually intended to improve quantification, the IPL is intended to transmit particles up to 2.5 μm into the instrument, not to improve quantification.

**L66:**

> We add "improved quantification **of the PM$_{2.5}$ fraction**" to the line, to take into account the intention of using the PM$_{2.5}$ IPL as well.

**RC3-8.** L78: What do you mean with "variation of empirical NO$_x^+$ ratio for pAmN"?

The variation of empirical NO$_x^+$ ratio for pAmN refers to how the instrument response to NH$_4$NO$_3$ change over time in terms of NO$_2^+$ to NO$^+$ concentration ratio, which was obtained from calibration using NH$_4$NO$_3$ standard solution. Empirical here simply means "observation" or "experimental" from actual measurements. We will change the word choice "empirical" to "experimental".

L78 → **L85-86**:

> Second, we show the variation of  **experimental** NO$_x^+$ ratio for pAmN in CV-UMR-ToF-ACSM…

**RC3-9.** L100: I assume, 525 °C is the vaporizer temperature, right?

It is correct that 525 $^0$C refers to the temperature of the vaporizer. We updated the manuscript as follows:

L100 → **L107-108**:

> …and a capture vaporizer (CV, temperature ~525 $^0$C,…

**RC3-10.** L101: The vaporizer is centered on the vaporizer? Reword.

The authors revised the word choice as follows.

L101 → **L108:**

> … that has been aligned **with the particle beam**.

**RC3-11.** L112: According to the URG website, this cyclone has a PM$_{2.5}$ cut-off at 3 lpm flow rate.

Since we have a PM$_{2.5}$ aerodynamic lens and intend to turn the ACSM into PM$_{2.5}$ monitor, Aerodyne suggested to use a larger cyclone cut-off to ensure that all PM$_{2.5}$ fraction are sampled through the inlet (e.g. PM$_{10}$ cyclone). However, since we only have the PM$_{2.5}$ cut-off cyclone, we intentionally reduce the flow rate to ~2 L min$^{-1}$ to let in particles with larger sizes so that we do not lose particles that are right on the cut-off size. The flow rate we described in the manuscript corresponds to the flow rate we set up rather than the flow rate prescribed by URG.

**RC3-12.** L129-131: Why are most LODs lower for this instrument at 2 min averaging time, compared to the other one with 10 min averaging time? Do the instruments generally behave differently due to different measurement history?

The LODs are influenced by on the condition of the instrument, for instance the cleanliness of the vacuum chamber, and the voltage detector. ACSM-UU has been deployed for many years in Cabauw, a polluted rural site, while ACSM-RUG has been measuring a relatively cleaner air from a coastal site. Therefore, it may affect the background electric noise of each instrument and explain this performance difference.

**RC3-13.** L151: I suggest rewording to "… assumed to be exclusively of organic origin."

We modify this information as suggested:

L151 → **L158:**

> …of further fragmentation of fragments at m/z 30 and m/z 46 and assumed to be exclusively **of organic origin**.

**RC3-14.** L152: The "mass concentration of organic fragment at m/z 30 and m/z 46" does not sound correct. There is nothing like "a mass concentration" of individual ions (even though it is clear, what you really mean). I suggest rewording to something like "… the signal contribution of organic fragments at m/z 30 and m/z 46".

The same comment holds for Line 157.

We modify the sentence as follows.

L152 → **L159:**

> …the fragmentation table, with respect to the $NO_x^+$ species, is to predict the **signal contribution** of organic fragments at m/z 30 and m/z 46 based on the masses measured…

To make the language uniform, we also modify the term "mass concentration" or "concentration" to "signal contribution" or "signal" or "signal contribution" throughout the manuscript when talking about ions or fragments.

- L78 → **L85:** "varying composition to better calculate $NO^+$ and $NO_2^+$ **signal contributions**."
- L166 → **L206, Figure 1 caption:** "The left-hand panels show the best ODR fits (set to zero intercept) which are found in the **relationship** between the **signal contributions** of (a) frag_Org[30] vs frag_Org[29],…"
- L167-168 → **L176-177:** "Switching from SV to CV  **modifies** the **signal** ratio between organic and inorganic fragments at m/z 30 and m/z 46,…"
- L199-200 → **L213-214:** "The low **signal intensity** of both m/z 46 and m/z 45 may cause this underestimation and suggests that frag_Org[46] and frag_Org[45] may…"
- L228-230 → **L242-244:** "Directed into the center, the particles enter the CV cavity and experience augmented thermal decomposition, at which the $NO_2^+$ **signal intensity** is at its minimum, while the $NO^+$ **signal intensity** is highest. The $NO_2^+$ **signal intensity** increases as the particle beam moves closer to the edge of the vaporizer, …

- L269-270 → **L284-286:** "A negative (or below zero) $R_{pON}$ value is not chemically possible for **the**  ratio."
- L295-297 → **L313-315:** "This means the $NO_2^+$ **signal intensity** is regularly close to the detection limit, particularly when the total $pNO_3$ concentration is low. This behavior also leads to noisy $R_{obs}$, due to a computation of very low or negative $NO_2^+$ signals, poor baseline, or both.
- L299-303 → **L316-321:** "…, we could discard observed $NO_2^+$ **signal intensities** that are below the detection limit. …. Therefore, we use observed $NO^+$ **signals** as the filtering parameter. The $NO^+$ **signal** accounts for ~95% of the total concentration of $NO_3$ species … and thus is a good indicator of when both $NO^+$ and $NO_2^+$ **signals** are too uncertain."
- L304 → **L322:** "Eq. 4 describes the $NO^+$ **signal** limit ($C_{NO+,lim}$) which assures reliable separation…"
- L307 → **L324-325:** "The measured data points with observed $NO^+$ **signal intensity**…"
- L310 → **L327:** "On this basis, we recommend data pre-treatments by time averaging and data filtering using observed $NO^+$ **signal contribution**…"
- L315 → **L332:** "The **signal** limit is lower as the time resolution increases due…"
- L320 → **L337, Table 3:**
    - Caption: "Detection limits of $NO_2^+$ and **signal** limits for $NO^+$ across different time averaging…"
    - Table header: "**Signal intensity** ($\mu g\ m^{-3}$)"
- L339 → **L357:** "of 6 components that make up $NO^+$ and $NO_2^+$ **signals**…"

**RC3-15.** L160: "A multiplier … is added" sounds odd. Better "is included".

The authors revised the word choice as suggested.

L160 → **L166-167:**

> A multiplier ***a*** (positive or negative) is **included** if the addition or subtraction of the component is fractional."

**RC3-16.** L163: The a_Org[x] multiplier in Table 1 is potentially different for every instrument, depending on e.g. particle beam alignment or vaporizer temperature or instrument history and potentially also dependent on the type of organic aerosol measured (which might affect fragmentation patterns of the organics). This should be made clear. As it is written right now, it sounds that a general multiplier can be used.

We intend to show that it can be used for CV-UMR-ToF-ACSM instruments rather than SV instruments. However, it is correct as mentioned by the referee that it depends on the particle beam alignment, vaporizer temperature, and the type of organic aerosol. We expect that the variability in the fragmentation table training set would reflect differences between instruments, tuning, alignment, etc.

What is a "CV inlet"? In my understanding the CV is part of the analysis section of the ACSM and not part of the inlet system. Same: line 164.

We modify the sentence as follows:

L163-164 → **L169-173**:

> The fragmentation table developed in this paper, therefore, is applicable to  **a** CV-UMR aerosol mass spectrometer . **Because the training data set incorporated**

**multiple chamber and ambient measurements with different instruments, it should be applicable for a range of typical measurement configurations, but users should be aware of the potential effects of the instrument condition (e.g., vaporizer temperature, particle beam alignment, measurement history).**

We also combine the term SV with inlet elsewhere. Therefore, we update them accordingly:

- L164 → **L174**: "In the default fragmentation table (which was developed using an SV-**based instruments),** the signal…"
- L286-287 → **L304-305**: "…in CV-UMR-ToF-ACSM datasets is a greater challenge than with SV , due primarily…"

To specify the vaporizer temperature for the table, we include additional information in **L190-191**:

**These spectra were obtained using vaporizer temperature ranging from 525 to 600 ºC (see Table S1 and S2). Therefore, the revised fragmentation table should be valid for CV-based instruments run in this temperature range.**

**RC3-17.** L165/166: This sentence is not correct. It sounds like that a small fraction of the signal at m/z 29 has a relationship to m/z 30. This is not what the frag table means. Furthermore, the frag table does not deal with correlations but with relationships between m/z-related signals.

The authors agree with the statement from the referee. We update the manuscript as follows:

L165-166 → **L175-176**:

…and  **the relationship of organic signal at m/z 30 is found to be only 0.022 times the magnitude of organic signal at m/z 29.**

To harmonize the language, we also updated the term "correlation" or "correlated" when talking about signals from two m/z into "relationship" or "related", unless we are talking about correlating in the statistical sense, as follows:

- L149-150 → **L155-156:** "… in the vaporizer and ionizer can be **related** to one another …"
- L166 → **L206, Figure 1 caption**: "The left-hand panels show the best ODR fits (set to zero intercept) which are found in the **relationship** between the **signal contributions** of (a) frag_Org[30] vs frag_Org[29],…"

**RC3-18.**

L166: may be better "larger contributions of organic fragments at …".

L167-169: I would also argue that because of the greater $NO_3$ fragmentation in CV (and consequently smaller remaining $NO_2$ (m/z 46) signal fraction) the correction of m/z 46 for organics contributions is much more relevant.

The authors will complete this information as suggested by rearranging the sentences in the paragraph (L164-170 → **L174-180**):

In the default fragmentation table … Switching from SV to CV  **modifies** the **signal** ratio between organic and inorganic fragments at m/z 30 and m/z 46, … . It also leads to greater nitrate fragmentation and consequently smaller $NO_2^+$, which makes organic contribution at m/z 46 more

important. **For instance,** … in a semi-polluted biogenically-influenced air analyzed with an SV-HR-ToF-AMS.

**RC3-19**. L171: Why SOA? Does POA not contribute to m/z 30 and 46?

POA probably contributed to m/z 30 and m/z 46 as well. To make it more general, the author will replace SOA to organic aerosol (OA) in this line:

L171 → **L183-184:**

To make a revised fragmentation table applicable for general ambient **organic aerosol (OA) mixtures,** a variety…

We also noticed that we never explained the abbreviation of SOA in the manuscript. We add this to L195 → **L209**, where SOA is appearing for the first time:

…(SV and CV) for a dataset dominated by biogenic **secondary organic aerosol (SOA).**

**RC3-20.** L179: Table S1, third and fourth column and Table 1, fourth and fifth column.

The authors will complete this information as suggested:

L179 → **L192-193:**

Using these data, we determine the multipliers a used in a revised calculation of frag_Org[30] and frag_Org[46] (see **Table 1, fourth and fifth column and** Table S1, third and fourth column).

In Table 1, we also **remove the third row labelling (a)-(d)** in the updated manuscript since we refer the column by its position, and the reader may confuse them with the footnote (a) and (b) of the table.

**Table 1.** Excerpt of fragmentation table for Org and $NO_3$ species in $m/z$ 30 and $m/z$ 46. Second and third column shows entries originated from the default fragmentation table of Allan et al. (2004) (used in Tofware v3.3). Fourth and fifth column shows entries proposed to develop revised CV-UMR-ToF-ACSM fragmentation table in this study.

| $m/z$ | Allan et al. (2004), default fragmentation table | | Proposed for general CV-ToF-ACSM | |
|---|---|---|---|---|
| | Org | $NO_3$ | Org | $NO_3$ |
| 30 | $0.022 \cdot$ frag_Org[29] | [30], -frag_Org[30] | $a_{Org[30],[i]} \cdot$ frag_Org[$i$] [a] | [30], -frag_Org[30] |
| 46 | - | [46] | $a_{Org[46],[i]} \cdot$ frag_Org[$i$] [b] | [46], -frag_Org[46] |

$i$ represents UMR masses tested against $m/z$ 30 and $m/z$ 46 in this study, which includes frag_Org[29], frag_Org[42], frag_Org[43], and frag_Org[45]. See the list in the footnote of Table S3 of SI.

(a) $a_{Org[30],[i]}$ is the multiplier for frag_Org[30] component, obtained from the slope of ODR fit between frag_Org[30] and frag_Org[$i$].

(b) $a_{Org[46],[i]}$ is the multiplier for frag_Org[46] component, obtained from the slope of ODR fit between frag_Org[46] and frag_Org[$i$].

**RC3-21.** L179/180: Are really the whole mass spectra correlated - or not rather only the signals at those m/z which are under investigation here (e.g., m/z 29 and 30 as well as 42/43/45 and 46)?

In general, the whole mass spectra should be correlated (have relationship between masses), but not for every m/z. Since we intentionally investigate m/z that are important to $NO_3$ determination, we only include the aforementioned m/z.

**RC3-22.** L186/187: This is not true. Figure 1a and 1b show the correlations between m/z 30 and m/z 29 and between m/z 46 and m/z 45, which apparently are the best correlations of all correlations that

were tested. This Figure does not, however, show that these are the best correlations since the other ones are not shown here. (Same comment line 191/192).

We modify the text to avoid this misunderstanding:

L186-189 → **L201-203:**

> **It is found that** frag_Org[30] is best correlated with frag_Org[29] **(see Table S3),** where $a_{Org[30],[29]}$ = 0.311 ± 0.016 (mean ± uncertainty, $r^2$ = 0.88**, see Figure 1a).** On the other hand, frag_Org[46] **has the best correlation** with frag_Org[45] (see Table S3), where $a_{Org[46],[45]}$ = 0.305 ± 0.037 (mean ± uncertainty, $r^2$ = 0.43, see **Figure 1b**).

L191-194 → **L205-207:**

> We apply these new multiplier values to the full dataset and compare the results with those from multipliers described in … . The result suggests that the multiplier $a_{Org[46],[45]}$ determined here gives the best predicted frag_Org[46] over multipliers from other studies ( ).

We also made a small update on Figure 1 (L166 → L206). The header text "total frag_Org[x] vs total frag_Org[i]" above plot (a)-(b) is updated into "**Correlating total frag_Org[x] vs total frag_Org[i]**" to describe better what is done to these plots.

**RC3-23.** L205ff (Section 3.2): Are all these results in this section generated also with a CV-ACSM or was a different instrument used in these studies?

We used the same dataset from CV spectra (25 CV-HR-AMS and 6 CV-UMR-ACSM) that are used in the previous section, except that we only take the chamber experiments where we know what the precursors are (and therefore the type of organic aerosol that may be formed).

The massive differences, especially for the $a_{Org[46],[45]}$, which span almost over an order of magnitude and which probably would directly translate in $NO_x$-ratios that span over a similar range, are a massive limitation for the presented method.

This must be discussed, and the resulting limitations of the method must be assessed. Is there a potential way out of this issue or does this mean that this method will not provide results better than the order of magnitude of NO3_Org and NO3_AmN for an unknown aerosol?

We agree with the referee's statement that the largest difference is coming from $a_{Org[46],[45]}$, since the values vary a lot depending of which data points/spectra are included in the ODR fitting. The smaller magnitude of m/z 46 and m/z 45 compared to m/z 30 and m/z 29 (thus more similar to noise) also contribute to the large range of $a_{Org[46],[45]}$ and small $r^2$ values. We have included the resulting uncertainties in the error propagation. The resulting limitations of the method has been discussed in RC1-4 as the sensitivity analysis and added to the manuscript.

**RC3-23.** L220: The $NO_x$ ratios for these two (nominally identical) instruments are very different from each other (more than a factor of two) while having very small individual uncertainties. What causes these huge differences? Different histories of the vaporizers? Different particle beam alignment? Different vaporizer temperatures? Different tuning of the instruments? All these influences have the potential to change over time.

I think it is crucial to know what causes such large changes in $NO_x$ ratio in order to have a robust method to calculate the different nitrate fractions.

We provided the $NO_x^+$ ratio for $NH_4NO_3$ ($R_{pAmN}$) values over the span of 1-2 years for both ACSM-UU and ACSM-RUG in Table S5 and observed that they are very stable (ran in the same configuration and vaporizer temperature).

We hypothesize that the $NO_x^+$ ratio for $NH_4NO_3$ ($R_{pAmN}$) of CV-ACSM instrument is unique to each instrument, even though their configurations are alike and tuned (e.g., aligned particle beam). A slight variation in the physical state of each component of the instrument may play role here, and we can only assume the "typical" range or value we expect for the $NO_x^+$ ratio of CV-ACSM instruments, which is 0.01-0.07 (mentioned in the manuscript in L234 → L248).

Another possibility is related to the different histories of the vaporizers as mentioned by the referee. This has been addressed in the response to RC3-12. ACSM-UU has been deployed for a long period in Cabauw, a polluted rural site, while ACSM-RUG has been measuring a relatively cleaner air from a coastal site compared to Cabauw. Therefore, it may affect how particles are collected and vaporized, which may explain this difference.

To conclude, the method should be robust as long as the empirical $R_{pAmN}$ falls within the typical $R_{pAmN}$ for CV instruments (0.01-0.07), and found to be stable over long time period.

**RC3-24**. L230: This sounds like there is a strong dependence of the $NO_x$ ratio on vaporizer temperature - is that the case?

The authors did not explore how large the dependence of the $NO_x^+$ ratio on vaporizer temperature. However, since we know that the particle bouncing and fragmentation inside the vaporizer depends on the architecture of the vaporizer (SV vs CV) and the temperature, it will be likely for $NO_x^+$ ratio to have dependence on the vaporizer temperature. Reports from Hu et al. (2017) and Hu et al. (2018a) have demonstrated this for both organic and inorganic nitrates. For $NH_4NO_3$, the $NO_x^+$ ratio varied within a small range and was an order of magnitude lower (0.015–0.04) in CV compared to SV, across a wide range of vaporizer temperature range (200 °C to 750 °C). In terms of organic aerosol, it was found that the fraction of signal at m/z < 50 for all OA types and oxidation levels is substantially larger (up to a factor of 2) for the CV compared to SV (vaporizer temperature range from 200 °C to 800 °C). Organic fragments (e.g., $C_xH_yO_z^+, C_xH_yON^+, C_xH_y^+$) are also found to be affected by the vaporizer temperature, which could also impact the organic nitrate $NO_x^+$ ratio since they are included in m/z 30 and 46 (see Figure R2).

We have taken into account the variability of the temperature into our revised fragmentation table by using CV spectra that were obtained at variable temperatures, listed in Table S1 and Table S2 in Supplementary Information. These spectra were recorded using vaporizer temperature ranging from 525-600 °C, close to the temperature suggested by Aerodyne for CV, ~550 °C. We addressed this information in the response to RC3-16.

**RC3-25**. L268, Figure 2: It would be helpful if the nomenclature used in the Figure and in the Figure caption would be the same. Furthermore, partially the description of the Figure does not agree with the Figure itself - e.g., that the "NOx+ ratio" is shown in panels c and d.

The authors noticed the mistakes and would like to thank the referee for pointing them out. The figure caption is changed as follows:

L268 → **L283, Figure 2 caption:**

> The time series of (a and b) $R_{obs}$, (c and d) ACSM species concentration (in $\mu g\ m^{-3}$, left bottom axis), and $f_{pON}$ (right bottom axis) of glyoxal+$NO_3$ chamber experiment at 15 min time averaging.

The UMR fragmentation table specific for glyoxal is used to obtain $C_{NO^+}$ and $C_{NO2^+}$. Panels (c) and (d) shows the progression of **$NO_3$ concentration**, **compared to panels (a) and (b) for** the $NO_x^+$ ratio during the formation of pON…

**RC3-26**. L271-272: Strictly, the range of ROR or the lower range of $R_{pON}$ should include values where $R_{pON}$ is zero (values below zero are physically not reasonable). Then there is an extremely large uncertainty in the determination of $R_{pON}$ with this method.

*RC3-37 (moved up here as the reply is related to RC3-26). L426: As this lower limit (for $R_{pON}$) was set arbitrarily, can you determine how the results would change if this lower limit would have been selected differently, e.g., lower by an order of magnitude?*

We do not include the value $R_{pON} = 0$ as the lower range since it would also be chemically inaccurate to have this value, implying that $NO_2^+ = 0$ (all $NO_2^+$ are fragmenting into $NO^+$). On top of that, RoR cannot be calculated with $R_{pON}$ being zero.

We have shown the sensitivity analysis of $R_{pON}$ to pON concentration using ambient pollution episode dataset presented RC1-4, Figure R3. The results suggest that as long as the $R_{pON}$ value is $\leq 10^{-3}$, the reported pON concentration is insignificantly different and therefore can be referred to as the lower limit of $R_{pON}$ and pON concentration. Therefore, we decided to use $R_{pON} = 0.0001$ instead of $R_{pON} = 0$. Our point of using $R_{pON} = 0.0001$ is not to dispose using $R_{pON} = 0$, but rather picking any number that is very small, approaching zero, that is still computable and physically reasonable.

On top of that, the uncertainty coming from $R_{pON}$ used in the analysis is expressed as the uncertainty range by using lower and upper limits of $R_{pON}$ in the final $f_{pON}$ calculation, as shown in Figure 7, Section 6.2. For instance, if we use $R_{pON} = 0$ as lower limit, the geometric mean of $R_{pON}$ summarized in Table 2 would be 0, regardless of the instrument, meaning we cannot approximate the uncertainty of $R_{pON}$ if we set the lower limit as 0.

**RC3-27.** L280ff: I am not very convinced about the robustness of the determined $R_{pON}$, calculated from an upper limit that is 35 and 72 times as large as the lower limit, which, on the other hand, is arbitrarily set to the same, very small, value. The consequences of this approach are two $R_{pON}$ values which are very similar to each other, suggesting a good agreement, while upper limits as well as $R_{pAmN}$ differ by a factor of two and there are 1.5 to 2 orders of magnitude in the range, determined for $R_{pON}$.

At least a reasonable analysis of the uncertainty for this approach is needed, that includes the uncertainty of this approach but also uncertainties due to the reasons which lie behind the differences (factor of 2) between the individual instruments. I would not be surprised if this results in an overall uncertainty in the order of a couple of hundred percent for the separation of AmN and ON.

The decision to choose a relatively small $R_{pON}$ value compared to $R_{pAmN}$ is supported by the new sensitivity analysis shown in RC1-4. To add, this choice is also similar to what Kiendler-Scharr et al. (2016) did when setting $R_{pON}$ value to 0.1 in their $NO_x^+$ ratio method (see in Figure S1 from the article, attached below). This number represents the minimum $NO_x^+$ ratio observed in the field data sets. Kiendler-Scharr et al. (2016) also mentioned how such low $NO_x^+$ ratios (for SV instrument in their case) were also detected in some data sets where $R_{pAmN}$ (there denoted as $R_{calib}$) was reported different. It means, no change in $R_{pON}$ regardless of change in $R_{pAmN}$ in different instruments, with estimated uncertainty of $\pm 20\%$.

[Figure]

**Figure S1.** Summary of the observed $NO_2^+/NO^+$ ratio for simulation chamber experiments studying SOA formation from the reaction of biogenic VOCs with $NO_3$ (Isoprene, Limonene, and β-Pinene), the fragmentation ratio of $NH_4NO_3$ in mixed laboratory generated aerosol ($NH_4NO_3$ and mixtures of $NH_4NO_3$ with $(NH_4)_2SO_4$ and glutaric acid) and the observed ratio of $NO_2^+/NO^+$ in Cabauw 2008. Lines indicate the ratios used in the calculation of the mass concentration of pOrgNO3 and the limit of detection (grey vertical line).

**RC3-28**. L292-294: These two sentences seem to contradict each other. How are the DL for the $NO_2^+$ and $NO^+$ provided? Since there is probably nothing like 0.044 μg m⁻³ of $NO_2^+$ ions anywhere, these DL only make sense if they are given with relation to ambient $NO_3$ concentrations. This, however, would only make sense if the different magnitudes of the two related signals (as mentioned in the following sentence) are already accounted for in the DL calculation.

I suggest rewording this paragraph.

The DL for $NO_2^+$ and $NO^+$ are obtained from the detection limits of m/z 46 and m/z 30 (mentioned in L119-120 and L130-131 → L126-127 and L137-138), while DL for $NO_3$ encompasses all m/z that are used to calculate the $NO_3$ species in Tofware. The signal m/z 46 and m/z 30 can be reported in either ion s⁻¹ or μg m⁻³ (processed by Tofware), but the authors chose the latter to show how it compares with the detection limit of total $NO_3$. Therefore, we add extra information to the manuscript regarding the $NO_3$ detection limit together with $NO_2^+$ and $NO^+$ detection limit to reduce the confusion.

L292-294 → **L310-312:**

> For instance, using the ACSM-UU, **the detection limit of $NO_2^+$ is comparable to the detection limit of $NO^+$ at pNO3 concentration near the detection limit of pNO3** ($C_{DL,NO2+}$ = 0.044 μg m⁻³; $C_{DL,NO+}$ = 0.066 μg m⁻³; **for $C_{DL,pNO3}$ = 0.075 μg m⁻³; all** in 10 min time resolution).

**RC3-29.** L302: As mentioned before, it seems not reasonable to use the term "$NO^+$ concentration" for the $NO_3$-related signal at m/z 30. There is nothing like an $NO^+$ ion concentration and definitely nothing in the order of a few ng/m³. Better use "$NO^+$ signal intensity"

Agreed. We have addressed this comment in the reply to RC3-14.

**RC3-30.** L305-307: I agree that with this criterion it is possible to obtain reliable $NO_3$ concentrations, however, if it is not possible to determine the ON fraction because the signal at m/z 46 is too low to reliably determine the ON contribution to it, how is it possible to calculate the fractional AmN contribution to it?

We agree that the sentence might be confusing, and therefore we decided to rewrite it as follows:

L305-307 → **L323-325:**

> We choose the larger $R_{pAmN}$ value, which is a less strict limit relative to $R_{pON}$ value, but still keeps any data with sufficiently good signal-to-noise ratio .

**RC3-31.** L332, Figure 3, caption: I wonder how this apportionment can be called "reliable" if for all concentrations the uncertainty range starts at a fraction of 0 and for almost all concentrations it ends at a fraction of 1 for $f_{pON}$. I would say that this means that the ON fraction of the nitrate is largely unknown.

Here, we understood that the uncertainty range mentioned by the referee refers to the colored shading of the moving average, which is the standard deviation.

The chemical coordinate plots shown in Figure 3 shows the statistics of ambient data measured in Cabauw (net 205 days of data). It is not directly related to the uncertainty of the measurements itself, since the data in each moving average point represents measurements at different time point (not a repeated measurement). The trace is the moving average of ON fraction, sorted by the total $NO_3$ concentration. It shows us that in average, with increasing nitrate concentration, we have the tendency to have lower ON fraction. This has been demonstrated in other studies, such as Day et al. (2022). Since this is an average, there are also periods where it behaves differently from this tendency (likely from a combination of real atmospheric composition differences and also instrument uncertainty). The value of treating the data as a chemical coordinate plot, averaging over a long period is that it allows for a robust characterization of the average trend, even using a method with substantial uncertainties. Importantly, the standard errors represent how well the averages are known. Their small uncertainty ranges support that the trend characterized is robust.

We added this information to the updated manuscript.

**L337, Figure 3, caption:**

> Figure 3. Chemical coordinate plots (a) between $R_{obs}$ against $C_{pNO3}$ in Cabauw **(net 205 days of data),** and…

**L342-345:**

> **The value of treating the data as a chemical coordinate plot is to allow for a robust characterization of the average trend, even using a method with substantial uncertainties. Importantly, the standard errors represent how well the averages are known. Their small uncertainty ranges support that the trend characterized is robust.**

**RC3-32.** L359: There is not "proportionality" observable in the respective plots. The $f_{pON}$ just increases with increasing Org fraction and the $f_{pAmN}$ increases with increasing $NH_4$ fraction, however the values are not proportional to each other.

We reword the sentence to depict what is described by the referee:

> We observe that $f_{pON}$ **increases with increasing** fraction of organic aerosol concentration …, whereas the $f_{pAmN}$ **increases with increasing fraction** of particulate ammonium concentration…

**RC3-33**. L370-371: How do we observe that this method is able to separate the ON and AmN contributions to total nitrate? Just because it produces results which are only during a fraction of the time chemically impossible? Is there any evidence that this separation reflects reality? E.g., why is for the autumn event (the only one with higher Organics than Nitrate concentrations) the average fractional ON contribution the largest of all four examples? Why does $R_{obs}$ not seem to reflect the ratio of ammonium to organics?

We find that it is interesting to discuss why some ambient nitrate episodes give such ON fraction, or how the total organics and ammonium relates to the pON/pAmN fraction.

However, we limit the scope of this paper describing the $NO_x^+$ ratio method to CV-ACSM data, and only showing how an apportionment of pON and pAmN in ambient nitrate episodes would look like, by showing Figure 5 and 6. We plan to write in detail a separate manuscript regarding possible physical/chemical processes involved in ambient ON formation in nitrate episodes, by exploiting the $NO_x^+$ ratio method and gas-phase measurements. We agree that these observations do not prove that the method works or exactly how well. Further application of the method to understanding chemical trends as well as future instrument/method comparisons will help to better validate and improve the application.

**RC3-34.** L383: How do you know that this RoR is correct for the type of ON, generated under these conditions. This might explain the differences in Figure 7c.

We do not know that this RoR is correct for the type of ON, since the RoR is generally assumed to apply to any organic nitrate compound. We do not have any strong reason to think that the RoR is dependent on the aerosol compositions, as has been shown in the large survey done with the SV-HR-AMS in Day et al. (2022). In that report, they concluded that there were no clear trends with composition. That is expected, given the tendency of $-NO_2$ to decompose with heating.

We do not have any reason to believe that this would be different for the CV, but we do not have as many measurements of oraganic nitrate with CV instruments. Because of this uncertainty, we use the upper limit, lower limit, and geometric mean of $R_{pON}$.

Even if we know the RoR, $R_{pON}$ calculated from that RoR will never be below zero or below the lower limit of $R_{pON}$, the value that we have shown in Figure 7c (in the preprint) as the lower whisker of $f_{pON}$ trace of CV-UMR measurements. With this value, we only have a small overlap to the value reported by the SV-AMS.

**RC3-35.** L385, Figure 7: Why was not the same averaging method used for the CV- and the SV-instruments?

The authors assume that the "averaging method" here refers to the time averaging used for the time series. For both instruments, 2 min averaging are used since the CV-ACSM was run in that time resolution for the campaign in order to capture more details in the chemistry, and we adjusted the SV-AMS to the same time resolution.

To clarify this, we updated the caption of Figure 7 as shown in RC1-4 as **Figure R4** (**Figure 8, L420,** in the updated manuscript) to explicitly mention the 2 min time averaging for each.

**RC3-36.** L410: A "successful separation" was shown. For ambient concentration levels, there is no indication that the calculated separation reflects the actual ambient separation of ammonium and organic nitrates. This, e.g., could be done using co-located HR-AMS measurements with SV.

We agree with this statement but unfortunately, we do not have any co-located HR-AMS measurements with SV for ambient measurements to be published. Therefore, as alternative, we rely on the co-located SV-HR-AMS and CV-UMR-ACSM measurements from the chamber experiment of CAINA campaign, as well as the newly added PMF analysis to the CV-UMR-ACSM data (see response in RC1-4) to show the successful separation.

We updated the manuscript as follows:

L410 → L477:

> We have  shown the separation of particulate ammonium nitrate (pAmN)…

**RC3-37.** L426: As this lower limit (for $R_{pON}$) was set arbitrarily, can you determine how the results would change if this lower limit would have been selected differently, e.g., lower by an order of magnitude?

We merged the response to this comment with RC3-26.

**RC3-38.** L444: It would be desirable if a statement about uncertainty and accuracy of this method would be clearly given: How well does it work for separation of AmN and ON in ambient measurements?

Since we only validate the method using chamber experiment, the only statement we can add is the assumption that it will also work properly for ambient measurements, with some precautions that a further work is needed to prove this statement.

**References**

[revised manuscript text omitted]